# RNA binding protein Caprin-2 is a pivotal regulator of the central osmotic defense response

Agnieszka Konopacka[1†], Mingkwan Greenwood[1], Su-Yi Loh[2], Julian Paton[3], David Murphy[1,2]*

[1]School of Clinical Sciences, University of Bristol, Bristol, United Kingdom; [2]Department of Physiology, Faculty of Medicine, University of Malaya, Kuala Lumpur, Malaysia; [3]School of Physiology and Pharmacology, University of Bristol, Bristol, United Kingdom

**Abstract** In response to an osmotic challenge, the synthesis of the antidiuretic hormone arginine vasopressin (AVP) increases in the hypothalamus, and this is accompanied by extension of the 3′ poly (A) tail of the AVP mRNA, and the up-regulation of the expression of RNA binding protein Caprin-2. Here we show that Caprin-2 binds to AVP mRNAs, and that lentiviral mediated shRNA knockdown of Caprin-2 in the osmotically stimulated hypothalamus shortens the AVP mRNA poly(A) tail at the same time as reducing transcript abundance. In a recapitulated in vitro system, we confirm that Caprin-2 over-expression enhances AVP mRNA abundance and poly(A) tail length. Importantly, we show that Caprin-2 knockdown in the hypothalamus decreases urine output and fluid intake, and increases urine osmolality, urine sodium concentration, and plasma AVP levels. Thus Caprin-2 controls physiological mechanisms that are essential for the body's response to osmotic stress.

*For correspondence: d.murphy@bristol.ac.uk

Present address: †Neuroscience and Pain Research Unit, Pfizer, Cambridge, United Kingdom

Competing interests: The authors declare that no competing interests exist.

## Introduction

The maintenance of salt and water homeostasis is a sine qua non condition for the survival of all living organisms (*Antunes-Rodrigues et al., 2004*). In mammals, this process is centrally regulated by the hypothalamo-neurohypophyseal system (HNS), the source of the antidiuretic neuropeptide hormone arginine vasopressin (AVP). The HNS consists of the large peptidergic magnocellular neurones (MCNs) of the hypothalamic supraoptic nuclei (SON) and paraventricular nuclei (PVN), the axons of which course though the internal zone of the median eminence and terminate on blood capillaries of the posterior pituitary (PP) gland (*Bargmann, 1966*; *Burbach et al., 2001*; *Antunes-Rodrigues et al., 2004*), a neuro–vascular interface through which the brain regulates peripheral organs in order to maintain homeostasis.

The biosynthesis of the AVP (*Brownstein et al., 1980*; *Burbach et al., 2001*) starts with the translation of AVP-encoding messenger ribonucleic acids (mRNAs) into a large precursor preprohormone that comprises a signal peptide (SP), the nine amino acid AVP moeity itself, the neurophysin II carrier molecule, and a C-terminal glycopeptide (copeptin) of unknown function. Delivery of AVP to the pituitary starts with the insertion of the preprohormone into the lumen of the endoplasmic reticulum with the removal of the SP. The resulting prohormone is folded then routed unidirectionally to the trans-Golgi network, where it is targeted specifically into the large dense core vesicle of the regulated secretory pathway. Whilst these granules are anterogradely transported some considerable distance towards the PP, the prohormone is processed to generate the biologically active, mature AVP, which is stored in PP axon terminals until mobilised for secretion into the systemic circulation. The rise in plasma osmolality that follows dehydration is detected by intrinsic MCN

**eLife digest** Cells are only able to work properly if they maintain a more or less constant balance of water and salts. In mammals, a hormone called arginine vasopressin regulates water and salt levels in the whole body. This hormone is made by cells in a region of the brain called the hypothalamus, and is then transported to the pituitary gland. When the level of water relative to the level of salts in the blood starts to drop (i.e., during dehydration), arginine vasopressin is released into the blood and travels to the kidneys where it acts as a signal to retain more water in the body.

However, if water levels continue to remain low, the stores of arginine vasopressin in the pituitary gland may run out and so more protein needs to be made in the hypothalamus. Like all proteins, arginine vasopressin is made by first copying a template encoded in a particular gene into a molecule called messenger ribonucleic acid (mRNA). During dehydration, the cells in the hypothalamus produce more of these corresponding mRNA molecules. Also, the mRNAs are slightly larger than normal because they have longer 'polyA tails' (structures added to the ends of all newly-made mRNAs). However, it was not clear how or why this happens.

Here, Konopacka et al. studied the production of arginine vasopressin in rats. The experiments show that a protein called Caprin-2 accumulates in hypothalamic neurons when rats are dehydrated. Furthermore, Caprin-2 is able to directly bind to the mRNA that encodes arginine vasopressin and is responsible for increasing the length of the polyA tail. To test whether this interaction is important for regulating the balance of water and salts, Konopacka et al. decreased the levels of Caprin-2 protein in the hypothalamus of live rats. When these rats became dehydrated, they had lower levels of the arginine vasopressin mRNA and these mRNAs had shorter polyA tails. Furthermore, the rats drank less water and urinated less than normal rats. Further experiments show that Caprin-2 helps to stabilize the structure of these mRNAs so that they accumulate in cells.

Together, Konopacka et al.'s findings show that Caprin-2 regulates the production of arginine vasopressin by interacting with and modifying its corresponding mRNA in the rat hypothalamus. The next challenge is to find out which other mRNAs in the hypothalamus are regulated by Caprin-2, and to determine their roles in the body.

mechanisms (*Bourque, 2008*) and by specialised osmoreceptive neurons in the circumventricular organs such as the subfornical organ, which provide excitatory inputs to shape the firing activity of MCNs for hormone secretion (*McKinley et al., 2004*). Upon release, AVP travels through the blood stream to specific receptor targets located in the kidney where it promotes water reabsorption in the collecting duct (*Breyer and Ando, 1994*), and sodium reabsorption in the thick ascending limb of the loop of Henle (*Ares et al., 2011*).

As a consequence of the depletion of pituitary stores that accompanies a chronic osmotic stimulation, there is a need to synthesize more AVP. This starts with an increase in transcription (*Murphy and Carter, 1990*), which results in an increase in the abundance of both the precursor heteronuclear RNA (*Kondo et al., 2004*) and the mature mRNA (*Sherman et al., 1986*). In addition, following the onset of an osmotic challenge, the AVP mRNA is subject to a curious post-transcriptional modification in the form of an increase in the length of the 3′ poly(A) tail (*Carrazana et al., 1988*; *Zingg et al., 1988*; *Carter and Murphy, 1989*; *Murphy and Carter, 1990*). Until this report, the regulation and physiologiocal function of this poly(A) tail length increase were not understood.

In order to better understand the network of plastic events in the osmotically challenged SON and PVN, and in an attempt to identify novel regulatory players, we (*Hindmarch et al., 2006*) and others (*Yue et al., 2006*) have used microarrays to ask how chronic osmotic stress evokes changes in the transcriptome. One of the genes found to be differentially up-regulated in the SON and PVN was cytoplasmic activation/proliferation-associated protein-2 (Caprin2) (*Mutsuga et al., 2004*; *Hindmarch et al., 2006*; *Yue et al., 2006*), otherwise known as C1q domain containing 1 (C1qdc1), EEG1 and RNA granule protein 140 (RNG140). In these various guises, Caprin2 has been implicated in the inhibition of cell growth (*Aerbajinai et al., 2004*), differentiation (*Aerbajinai et al., 2004*; *Lorén et al., 2009*), the enhancement of canonical Wnt signaling (*Ding et al., 2008*; *Miao et al., 2014*; *Flores and Zhong, 2015*), and, as an RNA-binding protein, the maintenance of the dendritic structure in the adult vertebrate brain (*Shiina and Tokunaga, 2010*).

We sought to understand the physiological role of Caprin2 in the function related plasticity exhibited following an osmotic challenge. Specifically, given that Caprin2 is an RNA binding protein, we tested the hypothesis that it may associate with AVP transcripts and hence mediate changes in poly(A) tail length and/or mRNA stability.

## Results

### Caprin-2 expression is up-regulated by osmotic stress in rat AVP neurones

Quantitative RT-PCR analysis (qRT-PCR) was used to demonstrate robust up-regulation of Caprin-2 mRNA expression in the rat PVN (*Figure 1A*) and SON (*Figure 1B*) following two different types of chronic osmotic stress—7 days of salt-loading (obligate consumption of 2% NaCl wt/vol in tap water) or 72 hr of dehydration (complete fluid deprivation). In parallel, AVP mRNA levels are significantly increased in both PVN and SON (*Figure 1A,B*). There was no significant change in Rpl19 levels (*Figure 1A,B*). Fluorescent immunostaining of brain slices revealed the presence of Caprin-2 in AVP MCNs in the PVN (*Figure 2A*) and SON (*Figure 2B*). Salt-loading (*Figure 2A,B*) and dehydration (not shown) both elicit a apparent increase in the intensity of staining. Mean Caprin-2 fluorescence in identified AVP cells also significantly increased (*Figure 2D*), whilst AVP signal in AVP cells significantly decreased (*Figure 2D*). We note that some MCNs express Caprin-2 but not AVP. These are possibly MCNs that express the closely related oxytocin neuropeptide hormone.

### Caprin-2 regulates fluid dynamics, plasma AVP and hypothalamic AVP mRNA levels in osmotically challenged rats

To study the physiological roles of Caprin-2 in the hypothalamus, we created a lentiviral vector (LV) expressing a specific Caprin-2 shRNA, which we used for Caprin-2 gene knockdown (Cap2 KD) in vivo. This shRNA was designed to target all known Caprin-2 transcript variants (see *Supplementary file 1*). As a control (Ctrl), we used an LV expressing a scrambled shRNA. Viruses with eGFP tags expressing the hairpins were delivered bilaterally into both the PVN and SON using stereotaxic brain surgery. First we showed, at the RNA level, that Cap2 KD results in a significant reduction in Caprin-2 mRNA levels in both the salt-loaded SON (*Figure 3A*) and PVN (*Figure 3A′*). Caprin-2 knockdown had no effect on GAPDH mRNA levels, which were similar in the Ctrl and Cap2 KD SON and PVN. The affinity of each LV for MCNs was confirmed by co-expression of eGFP with Caprin-2 and AVP (*Figure 3B*). We quantified the Caprin-2 fluorescent signal in eGFP-positive (transduced) compared to eGFP-negative (non-transduced) neurons in Ctrl and Cap2 KD rats. We found that in Cap2 KD rats, the Caprin-2 signal was significantly reduced in eGFP-positive neurons compared to eGFP-negative cells, whereas there was no significant difference in Caprin-2 signal between eGFP-negative and positive neurons in the Ctrl rats (*Figure 3C*).

In separate groups of animals, we then asked about the effects of Caprin-2 knockdown on fluid homeostasis in vivo. In euhydrated rats, Caprin-2 knockdown had no significant effect on any of the measured parameters. However, following salt-loading, Cap2 KD has profound effects. In Ctrl rats, as in naïve rats (*Greenwood et al., 2015*), salt loading resulted in increases in urine output and fluid intake that were both attenuated by Cap KD (*Figure 4A,B*; *Table 1*). Urine osmolality dropped significantly after SL in both Ctrl and Cap2 KD rats, but this was less pronounced in the latter animals (*Figure 4C*; *Table 1*). During salt-loading, there was a significant increase of urine [Na⁺] in both groups, however in Cap2 KD rats, urine [Na⁺] was significantly higher than in controls (*Figure 4D*; *Table 1*). At the end of the experiment, we measured plasma osmolality and AVP content. Caprin-2 knockdown had no significant effect on plasma osmolality ($308.7 \pm 1.49$ mOsmol/kg in Ctrl rats and $309.4 \pm 2.16$ mOsmol/kg in Cap2 KD rats, p = 0.78). However, plasma AVP levels in the Cap2 KD rats were significantly higher than in the Ctrl rats ($34.7 \pm 5.5$ pg/ml in Cap2 KD rats vs $21.6 \pm 2.8$ pg/ml in Ctrl rats; $p \leq 0.05$) (*Figure 4E*). We then used qRT-PCR to ask if the levels of AVP transcripts in the hypothalamus were changed following Cap2 KD. Paradoxically, in contrast to the increase in plasma AVP, we found that Caprin-2 mRNA knockdown (*Figure 3A,A′*) was accompanied by a significant decrease in AVP mRNA levels in SON (*Figure 4F*) and PVN (*Figure 4F′*). Note that Caprin-2 knockdown had no significant effect on food intake or body weight (not shown).

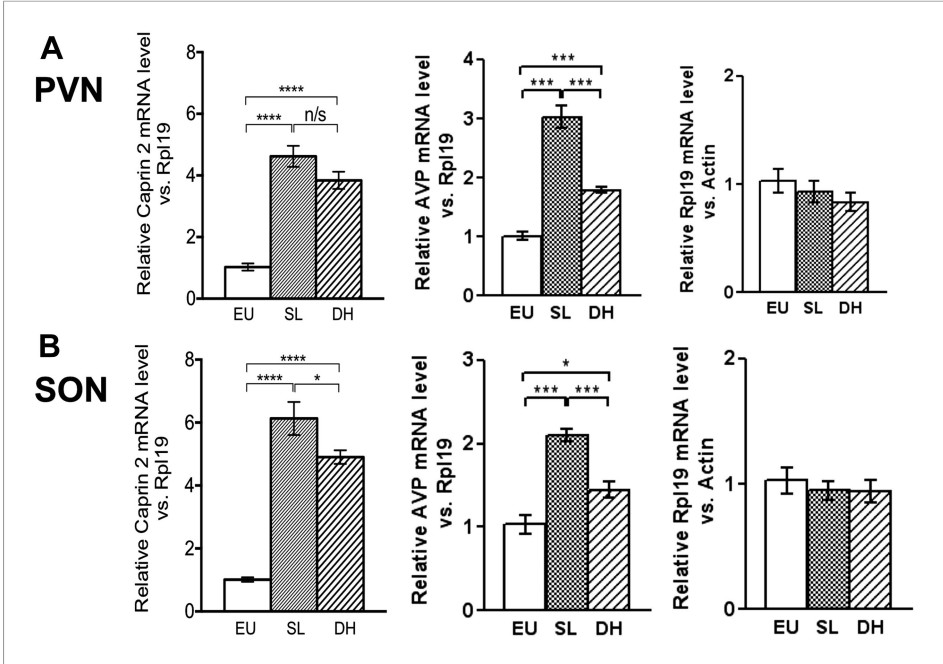

**Figure 1**. Caprin-2 messenger ribonucleic acid (mRNA) expression in the rat paraventricular nuclei (PVN) and supraoptic nuclei (SON) increases following a chronic osmotic stimulus. Quantitative RT-PCR analysis (qRT-PCR) analysis of Caprin-2 mRNA expression in the PVN (**A**) and SON (**B**) of euhydrated (EU), salt-loaded (SL) and dehydrated (DH) rats. *$p \leq 0.05$, **$p \leq 0.01$, ***$p \leq 0.001$, ****$p \leq 0.0001$, n = 5, One-way ANOVA with Sidak's post-hoc test. Compared with euhydrated (EU) rats, both SL and DH resulted in significant up-regulation of Caprin-2 mRNA in the PVN and SON. SL and DH significantly increased Caprin-2 mRNA levels in the PVN (euhydrated PVN, 1.03 ± 0.04; salt-loaded PVN, 4.62 ± 0.34; dehydrated PVN, 3.84 ± 0.28; n = 5; $p \leq 0.0001$ salt-loaded and dehydrated vs euhydrated). In the SON, SL further increased the Caprin-2 mRNA levels compared to DH ($p < 0.05$) and SON (euhydrated SON, 1.01 ± 0.07; salt-loaded SON, 6.13 ± 0.53; dehydrated SON, 4.9 ± 0.22 $p \leq 0.0001$ salt-loaded and dehydrated vs euhydrated, salt-loaded vs dehydrated). In parallel, arginine vasopressin (AVP) mRNA levels are significantly increased in both PVN (euhydrated PVN, 1.01 ± 0.07; salt-loaded PVN, 3.03 ± 0.18; dehydrated PVN, 1.79 ± 0.05; n = 6; $p \leq 0.001$ salt-loaded and dehydrated vs euhydrated, salt-loaded vs dehydrated) and SON (euhydrated SON, 1.03 ± 0.11; salt-loaded, SON 2.10 ± 0.07; dehydrated SON, 1.45 ± 0.09 $p \leq 0.001$ salt-loaded vs euhydrated, $p \leq 0.05$ dehydrated vs euhydrated, and $p \leq 0.001$ salt-loaded vs dehydrated). Rpl19 levels are unchanged in both the PVN (euhydrated PVN, 1.03 ± 0.11; salt-loaded PVN, 0.93 ± 0.10; dehydrated PVN, 0.83 ± 0.09, *ns*) and SON (euhydrated SON, 1.03 ± 0.10; salt-loaded SON, 0.95 ± 0.08; dehydrated SON 0.94 ± 0.09, *ns*).

## Caprin-2 protein binds to the AVP mRNA

Previous studies have shown that Caprin-2 is an RNA binding protein (*Shiina and Tokunaga, 2010*). We thus tested the hypothesis that in the rat SON and PVN, Caprin-2 might bind to the AVP mRNA. We performed an RNA immunoprecipitation assay on extracts from the SON and PVN of EU and SL rats using anti-Caprin-2 antibodies, followed by qRT-PCR (*Figure 5A*). The level of Caprin-2-AVP mRNA binding was quantified by comparing it to the signal detected with the non-specific binding control IgG performed simultaneously for each individual sample. In all samples incubated with Caprin-2 antibody, we found AVP mRNA at levels 1–2 orders of magnitude higher than in samples incubated with non-specific IgG. AVP mRNA levels in the Caprin-2-enriched extracts from EU and SL SON were respectively 49.44 ± 10.77 (n = 5, p = 0.002) and 23.77 ± 4.22 (n = 5, p = 0.0006) times higher, as compared to extracts incubated with non-specific IgG. In the EU and SL PVN, these values were respectively, 91.92 ± 24.25 (n = 4, p = 0.0037) and 108 ± 11.74 (n = 5, $p \leq 0.0001$). Binding to the control Rpl19 mRNA was negligible (not shown).

To examine the effect of salt loading on the amount of Caprin-2- bound AVP transcripts, we quantified Caprin-2-bound fractions in the SL rats relative to samples from the EU rats. The results showed significantly higher levels of AVP mRNA bound to Caprin-2 in the SL PVN and the similar trend

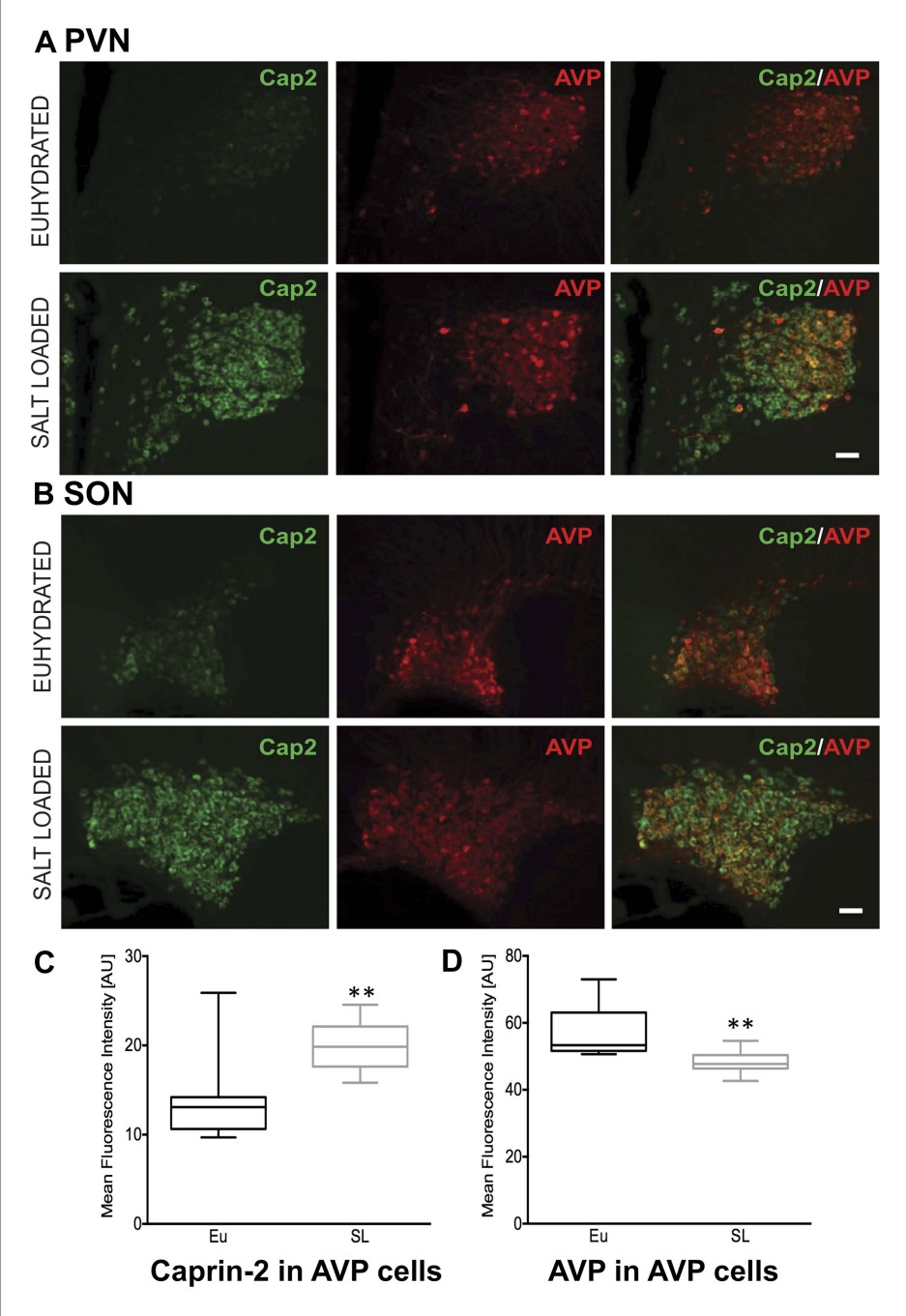

**Figure 2**. Caprin-2 protein expression in the rat PVN and SON. Immunohistochemical analysis of Caprin-2 protein expression (green) in the rat PVN (**A**) and SON (**B**) in euhydrated and salt-loaded rats; co-localization with AVP-neurophysin II (AVP; red). Scale bar 100 μm. (**C**) Quantification of fluorescence signals for Caprin-2 and (**D**) AVP in AVP cells. Mean fluorescence intensities were quantified in 10 SON wide field fluorescence microscope images acquired from 3 rats. AVP signal was selected (above the same threshold for EU and SL samples) and mean fluorescence intensity was measured for Caprin 2 and AVP signal. Each result was corrected by subtraction of mean background intensities for each channel. Salt-loading (SL) significantly increases Caprin-2 signal in AVP cells (euhydrated, 13.68 ± 1.47; salt-loaded, 19.92 ± 0.86, p = 0.0018), but results in a significant decrease of AVP in AVP cells (euhydrated, 56.94 ± 2.439; salt-loaded 48.3 ± 1.05, p = 0.0044). **p ≤ 0.01.

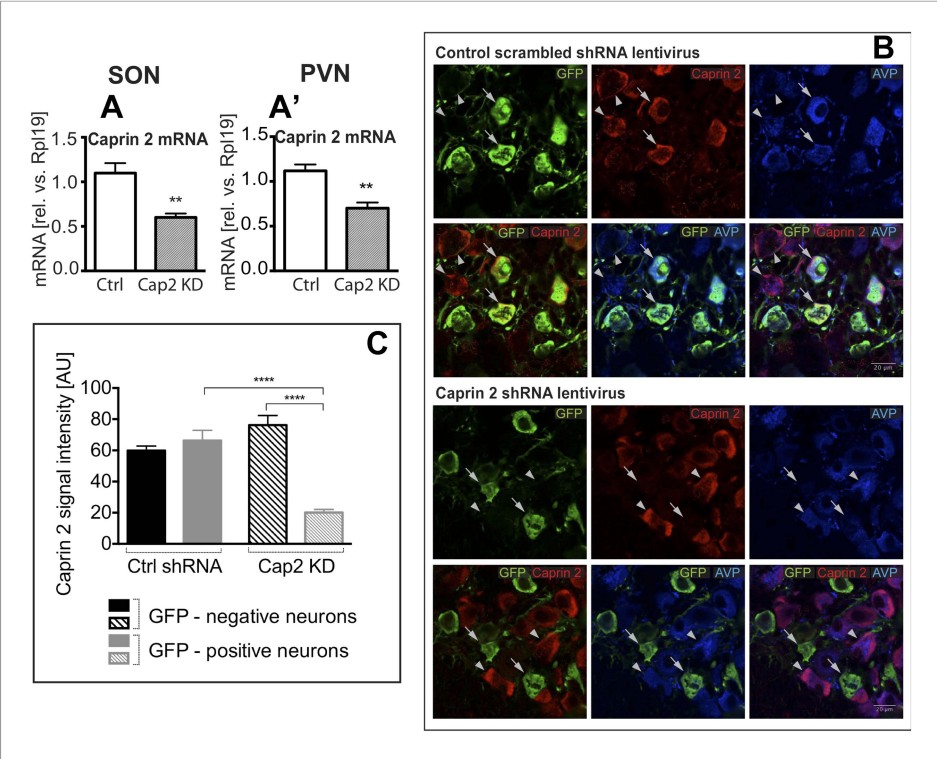

**Figure 3.** Lentivirus-mediated Caprin-2 shRNA knockdown in the SON and PVN. qRT-PCR analysis of the effect of Caprin-2 knockdown in the SON (**A**) and PVN (**A'**) on Caprin-2 mRNA levels (1.10 ± 0.11 vs 0.60 ± 0.04 in the Ctrl, n = 18, and Cap2 KD, n = 11, SON, p = 0.0023; 1.03 ± 0.07 vs 0.64 ± 0.06 in the Ctrl, n = 16, and Cap2 KD, n = 7, PVN, p = 0.0017). GAPDH mRNA levels are unchanged by Cap2 KD (1.03 ± 0.07 vs 1.04 ± 0.1, in the Ctrl, n = 18, and Cap2 KD, n = 11, SON, n.s.; 1.02 ± 0.05 vs 1.01 ± 0.1 in the Ctrl, n = 16, and Cap2 KD, n = 7, PVN, n.s.). (**B, C**) Immunohistochemistry-based quantification of Caprin-2 gene knockdown in magnocellular neurones (MCNs). (**B**) MCNs in the SON transduced with control scrambled shRNA and Caprin-2 shRNA lentiviruses visualized by immunostaining for eGFP (GFP, green) and co-localised with Caprin-2 (red) and AVP neurophysin II (AVP; blue). Full arrows—GFP positive cells. Arrow heads—GFP negative cells. Whilst the scrambled shRNA has no effect on Caprin-2 levels, in cells expressing the specific Caprin-2 shRNA, Caprin-2 expression is much reduced. (**C**) There was no significant difference in Caprin-2 signal between eGFP-negative and positive neurons in the Ctrl rats (respectively, 59.86 ± 2.89 and 66.25 ± 6.61; n = 18 and 23; n.s.) In contrast, in Cap2 KD SON, Caprin-2 signal was significantly reduced in eGFP-positive neurons compared to eGFP-negative cells (respectively, 20.16 ± 1.97 vs 76.19 ± 6.22; n = 15 and 26; p ≤ 0.0001). **p ≤ 0.01, ****p ≤ 0.0001.

was observed in the SON (*Figure 5B*). There was no change in the already low level of Caprin-2 binding to Rpl19 mRNA (*Figure 5C*), which does not change in abundance in the PVN or SON following an osmotic stimulus (*Figure 1A,B*).

## Caprin-2 regulates the length of AVP mRNA poly(A) tails in vivo

We and others have shown that osmotic stress results in the extension of the poly(A) tails of the AVP mRNA (*Carrazana et al., 1988*; *Zingg et al., 1988*; *Carter and Murphy, 1989*; *Murphy and Carter, 1990*). We thus hypothesized that Caprin-2 binding to the AVP mRNA might be involved in this process. Quantification of transcript sizes by Northern Blot revealed that Caprin-2 knockdown prevented the salt-loading-induced increase in AVP mRNA length in SON (*Figure 6A*) and PVN (*Figure 6A'*); the poly(A) returns to the length seen in euhydrated rats when Caprin-2 is knocked down. No changes were observed with the GAPDH mRNA (*Figure 6B,B'*). Removal of the poly(A) tails by hybridisation with oligo(dT) and incubation with RNase H reduced the size of the AVP mRNA, and removed differences between Ctrl and Cap2 KD animals (*Figure 6C,C'*).

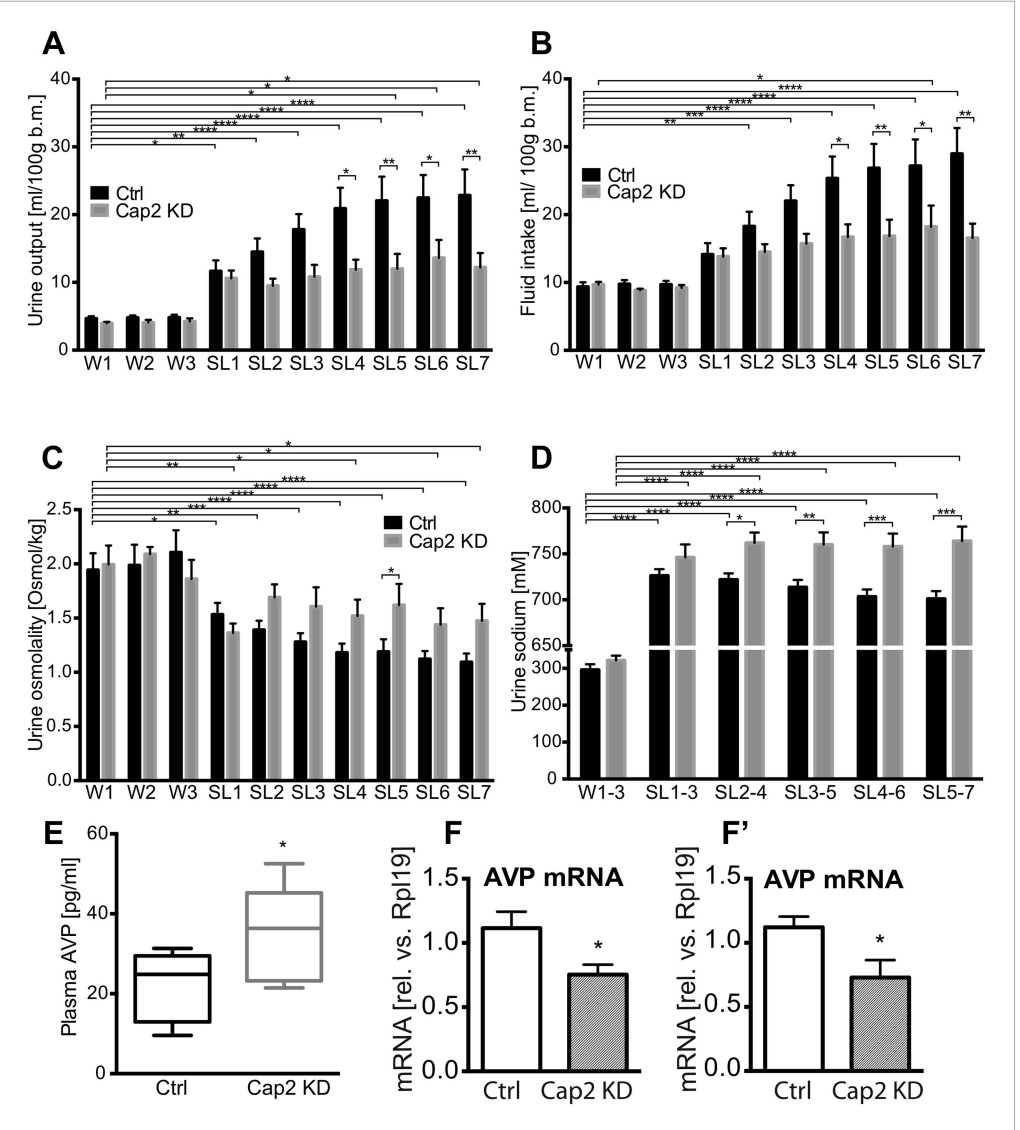

**Figure 4**. Physiological effects of Caprin-2 gene knockdown in euhydrated and salt-loaded rats. Urine output (**A**), fluid intake (**B**), urine osmolality (**C**) and urine sodium concentration (**D**) were measured in control, scrambled shRNA (Ctrl) and Caprin-2 shRNA lentivirus-injected (Cap2 KD) euhydrated (received water for 3 days, W1–3) and salt-loaded rats (received 2% wt/vol NaCl ad libitum for 7 days, SL1-7). Plasma AVP concentration (**E**) was measured after 7 days of SL, at the end of the experiment. *p ≤ 0.05, **p ≤ 0.01, ***p ≤ 0.001, ****p ≤ 0.0001, n = 9 (Ctrl) and 5 (Cap 2 KD). (**F**, **F'**) qRT-PCR analysis of the effects of Caprin-2 knockdown in the SON (**F**) and PVN (**F'**) on AVP mRNA levels (1.12 ± 0.13 vs 0.75 ± 0.08 for Ctrl, n = 18, and Cap2 KD, n = 11, SON, p = 0.047; 1.06 ± 0.08 vs 0.68 ± 0.13 for Ctrl, n = 16, and Cap2 KD, n = 7, PVN, p = 0.019). *p ≤ 0.05.

## Caprin-2 directly regulates AVP mRNA levels and the length of the poly (A) tail in vitro

We then developed a recapitulated in vitro system to ask if the effect of Caprin-2 on AVP mRNA abundance and poly(A) tail length was direct. We overexpressed the rat AVP structural gene and full-length rat Caprin-2 (*Supplementary file 1*) in HEK293T cells, both under the control of the heterologous CMV promoter (*Figure 7*). A vector expressing eGFP was used as a control. Western blotting showed robust expression of eGFP and Caprin-2 proteins (*Figure 7*). For knockdown studies, cells were additionally co-transfected with a Caprin-2 shRNA vector (overexpression control) or scrambled shRNA in place of the Caprin-2 shRNA (knockdown control).

**Table 1.** Urine output (A), fluid intake (B), Urine osmolality (C) and urine sodium concentration (D) in rats injected into the SON and PVN with either, control, scrambled shRNA or Caprin-2 shRNA, in euhydrated (water: W1–3) and salt-loading (SL 1–7) conditions

| | Scrambled shRNA | | | Caprin 2 shRNA | | |
|---|---|---|---|---|---|---|
| | Mean | SEM | N | Mean | SEM | N |
| A. Urine output (ml/100 g b.m.) | | | | | | |
| W1 | 4.661 | 0.344 | 9 | 3.954 | 0.222 | 5 |
| W2 | 4.784 | 0.339 | 9 | 4.048 | 0.423 | 5 |
| W3 | 4.803 | 0.413 | 9 | 4.230 | 0.461 | 5 |
| SL1 | 11.649 | 1.590 | 9 | 10.600 | 1.155 | 5 |
| SL2 | 14.482 | 2.005 | 9 | 9.488 | 1.062 | 5 |
| SL3 | 17.790 | 2.294 | 9 | 10.814 | 1.776 | 5 |
| SL4 | 20.884 | 3.086 | 9 | 11.860 | 1.485 | 5 |
| SL5 | 22.046 | 3.564 | 9 | 11.982 | 2.223 | 5 |
| SL6 | 22.473 | 3.367 | 9 | 13.598 | 2.666 | 5 |
| SL7 | 22.833 | 3.828 | 9 | 12.194 | 2.135 | 5 |
| B. Fluid intake (ml/100 g b.m.) | | | | | | |
| W1 | 9.353 | 0.668 | 9 | 9.674 | 0.421 | 5 |
| W2 | 9.752 | 0.607 | 9 | 8.832 | 0.251 | 5 |
| W3 | 9.684 | 0.542 | 9 | 9.203 | 0.425 | 5 |
| SL1 | 14.127 | 1.665 | 9 | 13.806 | 1.224 | 5 |
| SL2 | 18.270 | 2.145 | 9 | 14.483 | 1.156 | 5 |
| SL3 | 22.002 | 2.325 | 9 | 15.707 | 1.460 | 5 |
| SL4 | 25.351 | 3.211 | 9 | 16.680 | 1.886 | 5 |
| SL5 | 26.842 | 3.580 | 9 | 16.818 | 2.437 | 5 |
| SL6 | 27.166 | 3.912 | 9 | 18.209 | 3.124 | 5 |
| SL7 | 28.952 | 3.805 | 9 | 16.554 | 2.108 | 5 |
| C. Urine osmolality (mOsmol/kg) | | | | | | |
| W1 | 1943.333 | 154.937 | 9 | 1994.000 | 175.516 | 5 |
| W2 | 1986.667 | 190.343 | 9 | 2090.000 | 65.651 | 5 |
| W3 | 2105.556 | 204.702 | 9 | 1860.000 | 176.437 | 5 |
| SL1 | 1532.222 | 108.189 | 9 | 1362.000 | 87.772 | 5 |
| SL2 | 1390.000 | 85.261 | 9 | 1690.000 | 120.831 | 5 |
| SL3 | 1280.000 | 80.035 | 9 | 1606.000 | 178.342 | 5 |
| SL4 | 1181.111 | 83.773 | 9 | 1520.000 | 149.767 | 5 |
| SL5 | 1188.889 | 115.548 | 9 | 1620.000 | 195.090 | 5 |
| SL6 | 1121.111 | 75.064 | 9 | 1438.000 | 152.918 | 5 |
| SL7 | 1093.333 | 79.861 | 9 | 1474.000 | 156.831 | 5 |
| D. Urine sodium (mM) | | | | | | |
| W1-3 | 295.929 | 15.179 | 25 | 321.306 | 13.196 | 15 |
| SL1-3 | 726.221 | 7.102 | 27 | 746.221 | 14.048 | 15 |
| SL2-4 | 721.850 | 6.852 | 27 | 761.959 | 11.233 | 15 |
| SL3-5 | 713.653 | 7.925 | 27 | 759.992 | 13.389 | 15 |
| SL4-6 | 703.270 | 7.888 | 27 | 758.025 | 14.093 | 15 |
| SL5-7 | 701.085 | 8.319 | 27 | 763.926 | 15.838 | 15 |

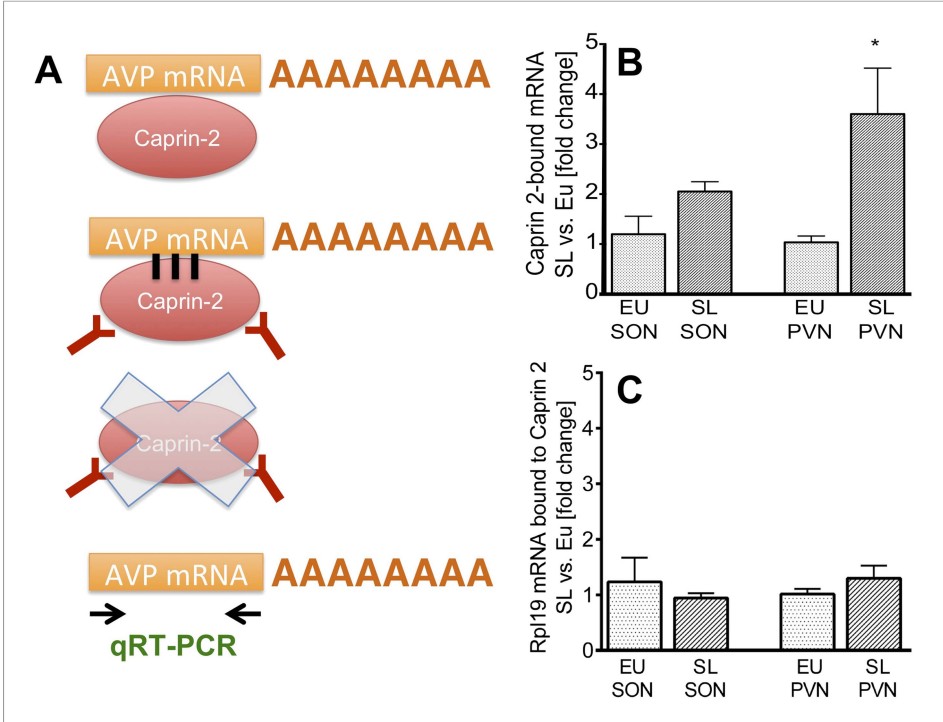

**Figure 5**. Caprin-2 binds to the AVP mRNA in the SON and PVN. Binding of AVP by Caprin-2 protein in the SON and PVN of euhydrated (EU) and salt-loaded (SL) rats determined by RNA immunoprecipitation assay. (**A**) In the RNA immunoprecipitation assay, SON or PVN tissue punches from EU or SL rats were first exposed to formaldehyde in order to covalently cross-link RNA with associated proteins. Cell extracts were then incubated with antibodies recognizing Caprin-2. Following immunoprecipitation, and hence enrichment of specific complexes, cross-links were reversed and extracted RNA was subject to qRT-PCR to detect AVP mRNA sequences. (**B**) Effects of salt-loading on the amount of Caprin-2 binding to AVP mRNA in the SON (1.2 ± 0.36 EU vs 2.05 ± 0.20 SL, n = 5, p = 0.072) and PVN (1.03 ± 0.13 EU vs 3.6 ± 0.92; n = 5, p = 0.0243). (**C**) Salt-loading has no effect on the amount of Caprin-2 binding to Rpl19 mRNA in the SON and PVN. *p ≤ 0.05.

Firstly, we showed using qRT-PCR that Caprin-2 knockdown significantly reduced Caprin-2 mRNA level, compared to the respective control (*Figure 8A*). Further, we found that the decrease in Caprin-2 mRNA abundance was accompanied by a 24% drop in AVP mRNA levels (*Figure 8B*). In contrast, overexpression of Caprin-2 resulted in a significant increase in AVP mRNA abundance (*Figure 8C*).

Northern blot analysis demonstrated that Caprin-2 knockdown significantly decreased the length of the AVP mRNA (*Figure 9A,E*). Removal of poly(A) tails reduced the size of the AVP mRNAs from Ctrl and Cap2 KD cells to the same length (*Figure 9A,E*). In contrast, Caprin-2 overexpression significantly increased the length of the AVP mRNA (*Figure 9B,F*). Removal of the poly(A) tails by hybridisation with oligo(dT) and incubation with RNase H reduced the size of the AVP mRNAs from all samples to the same length (*Figure 9B,F*). Neither, Caprin-2 overexpression nor Caprin-2 knockdown had an effect on the length of the GAPDH mRNA (*Figure 9C–F*).

## Discussion

We provide the first evidence that RNA binding protein Caprin-2 plays a critical role in mediating brain responses to osmotic stress. We show that Caprin-2 is expressed in AVP MCNs in the SON and PVN and that Caprin-2 expression in these neurons increases during osmotic stress. Importantly, we demonstrate that Caprin-2 knockdown in the SON and PVN disrupts physiological osmoregulatory mechanisms. Normally, chronic salt loading leads to a gradual increase of urine output and fluid intake. Loss of Caprin-2 significantly reduces the salt loading-induced urine output and fluid intake and increases urine osmolality, urine sodium concentration and plasma AVP levels.

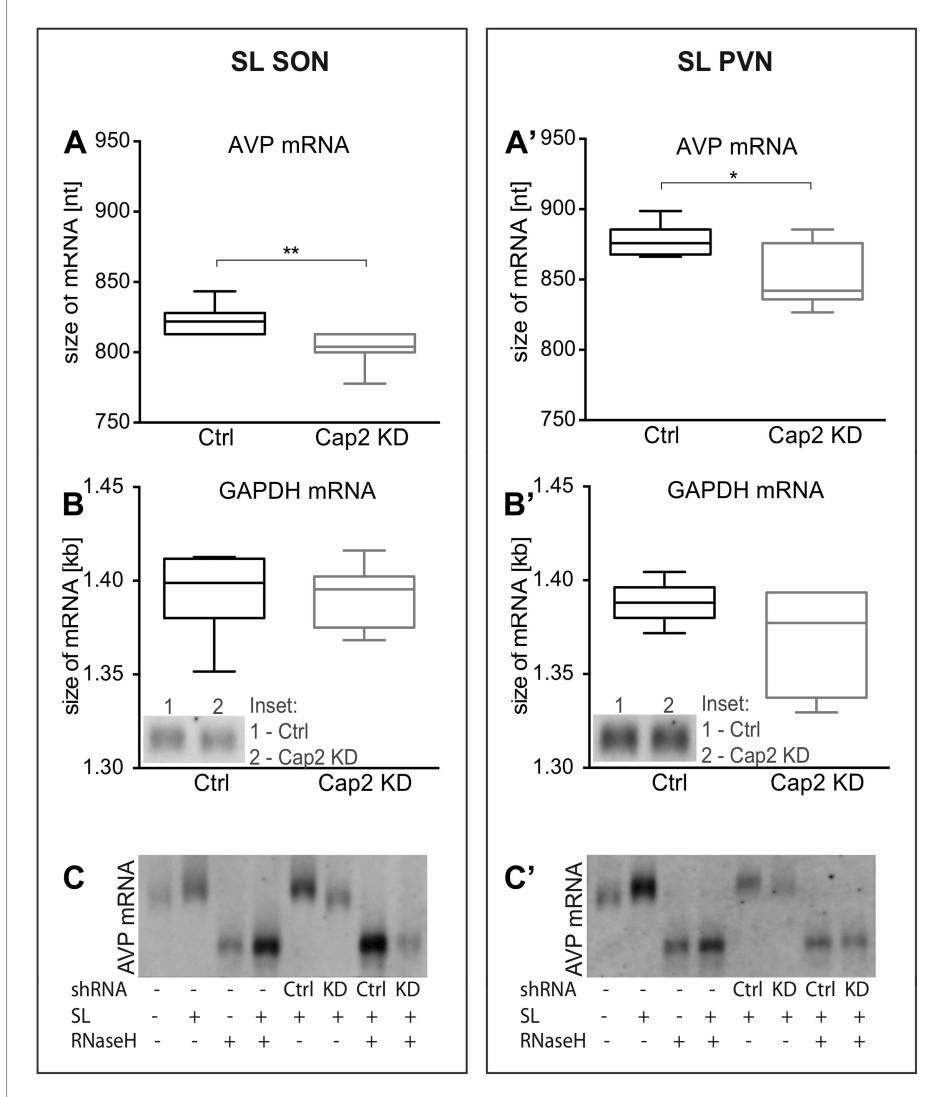

**Figure 6**. In vivo effects of Caprin-2 gene knockdown on the length of the AVP mRNA poly(A) tail. AVP mRNAs were analyzed by Northern Blotting. Quantification results of AVP (**A**, **A'**) and GAPDH mRNA (**B**, **B'**) in samples obtained from the SON (**A**, **B**) and PVN (**A'**, **B'**) of SL rats injected with lentivirus with either, scrambled shRNA (Ctrl) or Caprin-2 shRNA (Cap2 KD). Insets on the plots with GAPDH mRNA represent typical images. Representative images of AVP mRNA before and after removing poly(A) tails (RNaseH) in the SON (**C**) and PVN (**C'**) of euhydrated (–) and salt-loaded (SL) rats, as well as in salt-loaded rats injected with scrambled (Scr shRNA) or Caprin-2 (Cap2 shRNA) shRNA. Quantification of transcript sizes revealed that Caprin-2 knockdown prevented the SL-induced increase of the AVP mRNA length, reducing it from 822.7 ± 3.76 nt. (n = 8, Ctrl) to 801.7 ± 4.47 nt. (n = 7, Cap2 KD) (p = 0.0031) in the SON, and from 877.7 ± 4.77 nt. (n = 6, Ctrl) to 851.3 ± 9.23 nt. (n = 6, Cap2 KD) in the PVN (p = 0.0296). No size changes were observed with the GAPDH mRNA (Ctrl SON, n = 8, 1393 ± 7.55 nt. vs Cap2 KD, n = 7, 1392 ± 6.12 nt., n.s.; Ctrl PVN, n = 6, 1388 ± 4.65 nt. vs Cap2 KD, n = 6, 1368 ± 11.24 nt., n = 6, n.s.). *p ≤ 0.05, **p ≤ 0.01.

We then explored the molecular mechanisms of Caprin-2 action in homeostatic osmoregulatory hypothalamic neuronal circuits. Caprin-2 has recently been identified in vitro as an RNA binding protein RNG140 (*Shiina and Tokunaga, 2010*). However, up until now, no specific RNA binding partners of Caprin-2 have been identified. Here we show that Caprin-2 binds to the AVP mRNA in the SON and PVN of EU and SL rats in vivo, and, in doing so, modulates both its abundance and poly(A) tail length.

By stimulating water retention at the level of the kidneys (*Breyer and Ando, 1994*), AVP plays a central role in the maintenance of cardiovascular homeostasis, particularly blood volume and

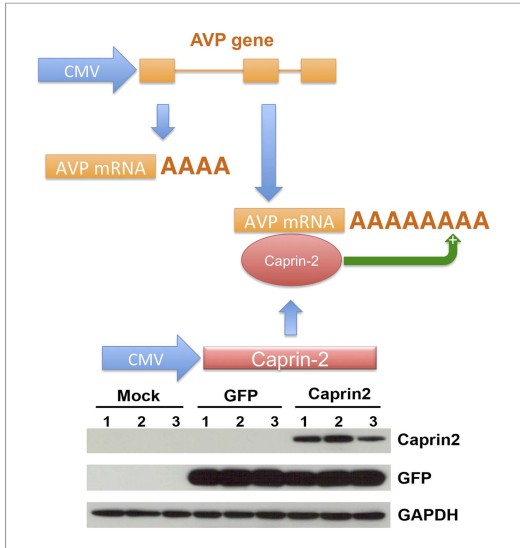

**Figure 7**. A recapitulated in vitro system for examining effect of Caprin-2 on the metabolism of the AVP mRNA. In order to further examine to role of Caprin-2 in AVP mRNA metabolism, we developed a recapitulated in vitro system. We overexpressed the rat AVP structural gene and full-length rat Caprin-2 in HEK293T cells, both under the control of the heterologous CMV promoter. A vector expressing eGFP was used as a control. Western blotting of triplicate independent samples showing robust expression of eGFP and Caprin-2 proteins in transduced cells.

osmolality (*Antunes-Rodrigues et al., 2004*). The increase in AVP biosynthesis in MCNs that follows osmotic stimulation is accompanied by a plethora of changes in gene expression (*Mutsuga et al., 2004*; *Hindmarch et al., 2006*; *Yue et al., 2006*). One of the novel genes identified as such by transcriptome analysis was Caprin-2. Here we confirm the transcriptome data, and that Caprin-2 protein is up-regulated at the mRNA level by chronic hyperosmotic cues. Using double immunostaining we also provide evidence that Caprin-2 protein is present in the cytoplasm of VP MCNs, and that its expression increases during osmotic stress. Caprin-2 has been reported to play a role in several physiological processes, including differentiation, apoptosis in erythroblasts or lens fiber cells, and synaptic plasticity in the mouse cerebellum (*Aerbajinai et al., 2004*; *Ding et al., 2008*; *Lorén et al., 2009*; *Shiina and Tokunaga, 2010*; *Miao et al., 2014*; *Flores and Zhong, 2015*). Although expression of Caprin-2 in the HNS has been reported (*Mutsuga et al., 2004*), its role in the maintenance of fluid homeostasis had not previously been addressed. We therefore investigated the physiological consequences of Caprin-2 gene knockdown using virally delivered specific shRNAs in the SON and PVN in vivo. The Caprin-2 shRNA had no effect on water homeostasis in euhydrated rats. In naïve rats (*Greenwood et al., 2015*), and in rats transduced with a scrambled control shRNA, as shown here, salt loading results in production of large amounts of dilute urine and excessive fluid intake. However, compared to the control animals, salt loaded Caprin-2 knockdown rats only minimally increased their urine production and they drank less fluid. Urine osmolality and sodium levels were also higher in Caprin-2 knockdown rats. These observations may be explained by water retention resulting from the elevated plasma AVP levels found in the Caprin-2 shRNA-transduced rats. Our data suggest that Caprin-2 knockdown rats increase their ability to concentrate urine and so doing they prevent the decrease in sodium excretion (i.e., elevation of plasma sodium) resulting from lower urine output by producing more concentrated urine.

We then explored the molecular mechanisms of Caprin-2 action in the hypothalamus, based on our hypothesis that this RNA binding protein (*Shiina and Tokunaga, 2010*) might associate with the AVP mRNA. Here we show that Caprin-2 does indeed bind to AVP mRNAs in the SON and PVN in vivo. We then asked if Caprin-2 binding has any role in the alterations in AVP mRNA metabolism that occur as a consequence of the chronic osmotic stimuli of dehydration or salt-loading, namely an increase in abundance (*Sherman et al., 1986*; *Murphy and Carter, 1990*), and the unusual post-transcriptional modification of an increase in the length of the 3′ poly(A) tail (*Carrazana et al., 1988*; *Zingg et al., 1988*; *Carter and Murphy, 1989*; *Murphy and Carter, 1990*). Our in vivo studies show that knockdown of Caprin-2 in the rat SON and PVN prevented the salt-loading-induced increases in the abundance of the AVP mRNA, and the elongation of its poly(A) tail. In HEK293T cells transfected with a plasmid containing rat AVP genomic sequences under constitutive CMV promoter, concomitant overexpression of Caprin-2 increased AVP mRNA abundance, whilst addition of the Caprin-2 shRNA significantly reduced AVP mRNA levels. At the same time, Caprin-2 overexpression resulted in elongation of the poly(A) tail of the AVP mRNA, whilst Caprin-2 knockdown shortened the poly(A) tails of the AVP mRNA. The direct binding and positive regulation of AVP transcript levels suggest that Caprin-2 may be important for stabilization of AVP mRNAs during osmotic stress, in a similar way to another AVP mRNA binding protein, poly(A)-binding protein (PABP) (*Mohr et al., 2002*). It is thus

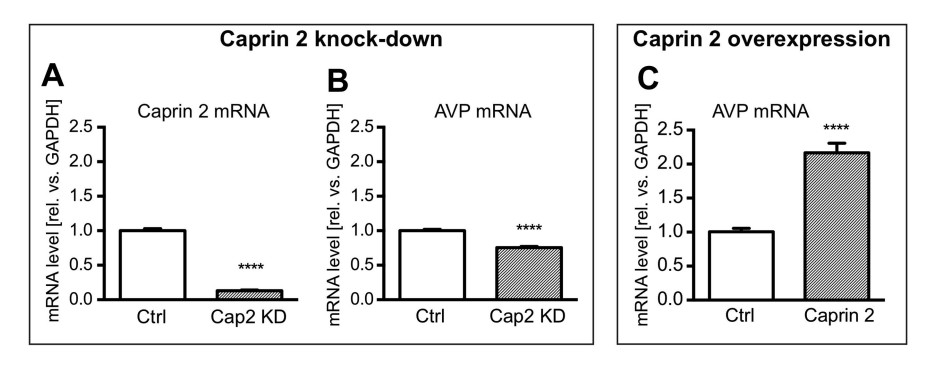

**Figure 8**. Effects of Caprin-2 overexpression or knockdown on AVP mRNA levels in vitro. For Caprin-2 knockdown experiments, cells were co-transfected with the CMV-driven rat Caprin-2 overexpression construct, the CMV-driven rat AVP structural gene, and either scrambled shRNA (Ctrl) or Caprin-2 shRNA (Cap2 KD) constructs. Caprin-2 knockdown (Cap2 KD) reduces (**A**) Caprin-2 (Ctrl, 1.00 ± 0.03 vs Cap2 KD, 0.13 ± 0.01, n = 5, p ≤ 0.0001) and (**B**) AVP (Ctrl, 1.00 ± 0.02 vs Cap2 KD, 0.76 ± 0.02; n = 5, p ≤ 0.0001) mRNA levels. In the Caprin-2 overexpression experiments, the CMV-driven rat AVP structural gene was co-transfected with either the CMV-driven rat-Caprin-2 overexpression construct (Cap2), or a control CMV-eGFP construct (Ctrl). (**C**) Caprin-2 overexpression increases the level of AVP mRNAs (Ctrl, 1.0 ± 0.05 vs Cap2, 2.17 ± 0.14, n = 5, p ≤ 0.0001). ****p ≤ 0.0001.

tempting to speculate that the two processes, increased poly(A) tail length and increased mRNA abundance, are consequential. An important role of poly(A) tail length in regulation of mRNA degradation and stability has been demonstrated for many transcripts (*Curinha et al., 2014*; *Zhang et al., 2014*). It is therefore possible that one of the functions of Caprin-2-mediated poly(A) tail length regulation is protection of AVP transcripts from degradation, which in turn would lead to the observed increase of AVP mRNA levels. Note that we are unable to distinguish between a process that promotes polyadenylation, or one that prevents deadenylation.

Unexpectedly, whilst Caprin-2 knockdown in the PVN and SON resulted in a decreased AVP mRNA levels in those structures, plasma AVP (peptide) levels increased. Although the precise mechanisms underlying this paradoxical effect require separate investigation, examination of the existing literature leads us to propose two possible hypotheses. Firstly, Caprin-2 binding to AVP mRNA might inhibit translation, a phenomenon observed as a consequence of other association mRNA-RNA-binding protein interactions, such as the binding of PABP to the AVP mRNA (*Mohr et al., 2002*; *Richter, 2008*). This is not necessarily in disagreement with the fact that Caprin-2 stimulates extension of poly(A) tail in AVP mRNAs—recent studies have revealed that a correlation between the length of poly(A) tail and translational activity is not always present (*Weill et al., 2012*; *Subtelny et al., 2014*).

It has been shown previously that recombinant Caprin-2 inhibits the translation of a luciferase mRNA in a cell-free rabbit reticulocyte lysate system (*Shiina and Tokunaga, 2010*). This is consistent with our observation that Caprin-2 knockdown in the SON and PVN increases circulating plasma AVP levels. It is conceivable that this effect results from the increased translation of AVP mRNA under conditions when Caprin-2 protein level is too low to suppress this process. Moreover, knockdown of Caprin-2, whilst increasing the translation of the AVP mRNA, might increase its turnover, and hence decreases steady state transcript levels and poly(A) tail length. We have previously examined the polysome distribution of the AVP mRNA in euhydrated and salt-loaded SON (*Murphy and Carter, 1990*), and showed no difference in the pattern association with heavy polysome fractions. However, the AVP mRNA is small, and subject to a high rate of translation, even in euhydrated animals. Thus, the rates of translation initiation, elongation and termination, none of which are directly measured by polysome analysis, especially when the size of the RNA limits ribosome number, could be influenced, positively or negatively, by poly(A) tail length. Quantification of Caprin-2 and AVP levels in AVP in the SON neurons (*Figure 2*) revealed that salt-loading elicits a significant increase in the former protein, but a significant decrease in the latter, consistent with Caprin-2 being an inhibitor of the translation of the AVP mRNA. However, it is important to point out the steady-state level of AVP in hypothalamic MCNs is not only determined by translation rate. As we describe in some detail in the Introduction,

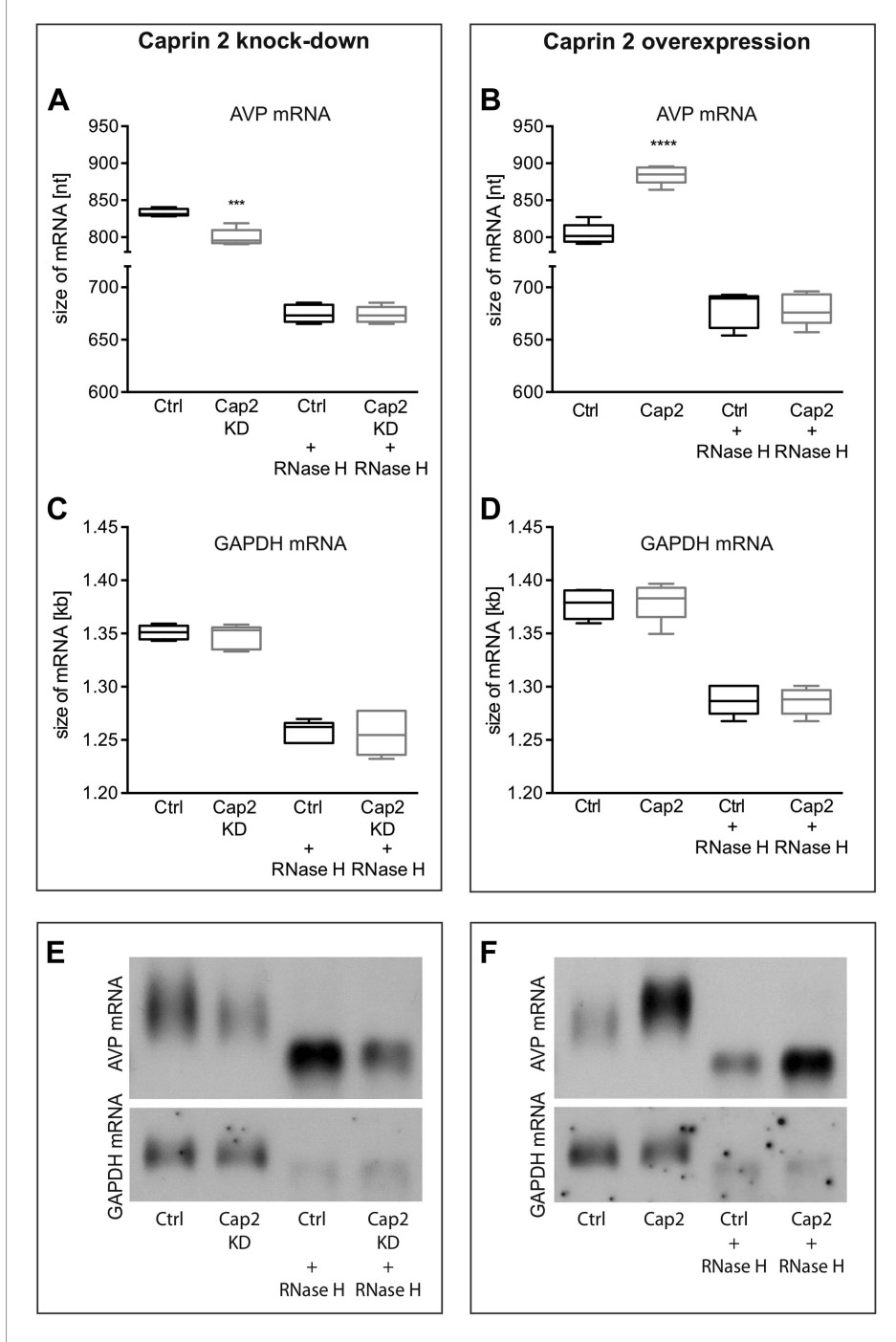

**Figure 9.** Effects of Caprin-2 overexpression or knockdown on AVP mRNA poly(A) tail length in vitro. Effects of Caprin-2 knockdown (Cap2 KD) (**A**) and overexpression (Cap2) (**B**) in HEK293 cells on the length of the poly(A) tail in AVP mRNA, analyzed by Northern Blotting. Knockdown reduces (Ctrl, 833.3 ± 2.21 nt. vs Cap2 KD, 799.6 ± 5.07 nt., n = 5, p = 0.0003) whilst overexpression increases (Ctrl, 804.4 ± 6.19 nt. vs Caprin-2, 884.3 ± 5.51 nt., n = 5; p ≤ 0.0001) the length of the AVP mRNA. This is due to modulation in the lenth of the poly(A) tail, when this is removed there are no differences between the groups (Ctrl, 674.8 ± 3.75 nt. vs Cap2 KD, 674 ± 3.48 nt., n = 5; n.s.; Ctrl, 679 ± 7.62 nt. vs Cap2, 678.9 ± 6.79 nt., n = 5; n.s.). Neither Caprin-2 knockdown (**C**) (Ctrl, 1351 ± 2.94 nt., vs Cap2 KD, 1347 ± 4.98 nt., n = 5, n.s.) nor Caprin-2 overexpression (**D**) (Ctrl, 1377 ± 6.13 nt. vs Caprin-2, 1380 ± 8.05 nt., n = 5; n.s.) had any effect on the size of the GAPDH mRNA. Typical images representing the effects of Caprin-2 knockdown (**E**) and overexpression (**F**) on AVP mRNA in HEK293T cells before and after removing poly(A) tails (RNaseH) are shown. ****p ≤ 0.0001, ***p ≤ 0.0003.

the AVP precursor is subject to axonal transport from cell bodies in the hypothalamus to terminals in the PP, a process that is activated by the need to deliver mature peptide to the circulation following an osmotic stimulus. Thus, further studies are needed to examine the translation of the AVP mRNA under different physiological conditions.

A second, but not mutually exclusive hypothesis is that the effects that we observe after Caprin-2 knockdown could be modulated by changes in dendritic release of AVP. Dendritic release and autocrine or paracrine action of AVP has been well documented (*Ludwig and Stern, 2015*). AVP released from dendrites inhibits the electrical activity of AVP neurons, thus suppressing axonal release to the blood stream (*Ludwig and Stern, 2015*). Caprin-2 protein is present in the RNA granules localized in the rat neuronal dendrites (*Shiina and Tokunaga, 2010*). Therefore, it is possible that Caprin-2 knockdown results in a decrease of the synthesis and dendritic release of AVP, which in turn decreases AVP-mediated auto-inhibition of AVP neurons and leads to an increase of AVP release in the neurohypophysis and increased plasma concentration, as we observe after Caprin-2 knockdown in the SON and PVN.

In this report, we have shown that, during an osmotic stimulus, three things happen to the AVP mRNA in the SON and PVN due to the action of Caprin-2: the abundance of Caprin-2 protein/AVP mRNA complexes increases, the AVP mRNA poly(A) tail elongates, and there is an increase in steady state levels of the AVP mRNA. On the basis of these data, we propose a new mechanism mediating the regulation of AVP gene expression (*Figure 10*). Whilst it has previously been demonstrated that a chronic osmotic stimulus results in an increase in the transcription of the AVP gene (*Murphy and Carter, 1990*), the data presented here provides new evidence that post-transcriptional process, governed by Caprin-2, are also operative. Both the transcriptional and the post-transcriptional mechanisms would contribute to overall steady-state AVP mRNA abundance. However, it would appear that the increase in AVP mRNA mediated by Caprin-2 does not result in an increase in AVP

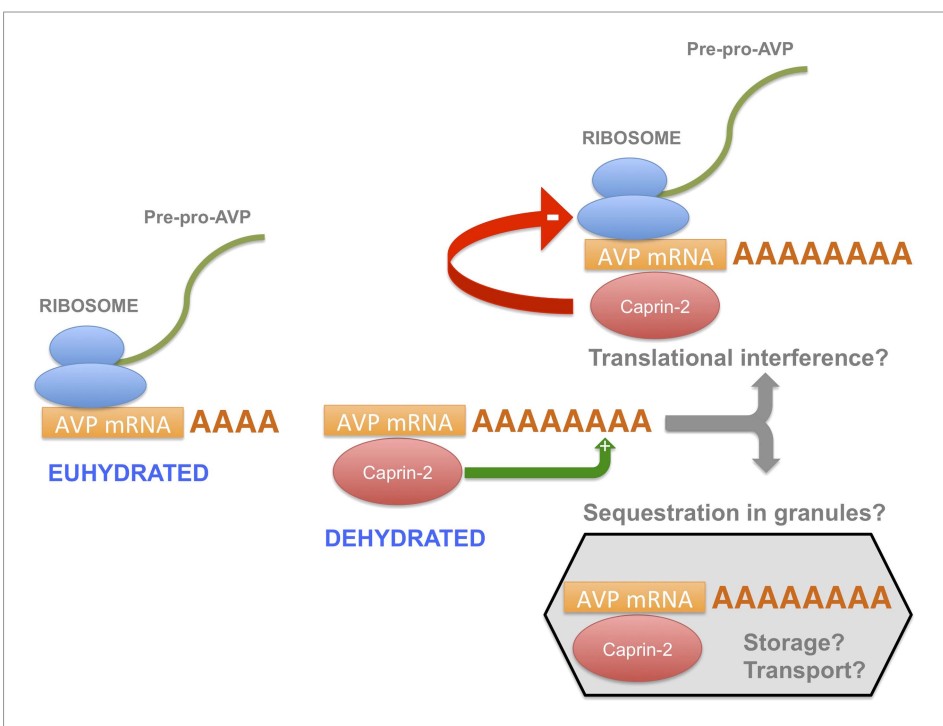

**Figure 10**. A model for the actions of Caprin-2. Caprin-2 expression increases in the rat SON and PVN following an osmotic stimulus. Caprin-2 binds to the AVP mRNA, and mediates an increase in the length of the poly(A) tail, either by stimulating further synthesis, or by inhibiting deadenylation. Caprin-2 also mediates an increase in VP mRNA abundance, possibly by increasing transcript stability. Paradoxically, we propose that Caprin-2 inhibits translation of the AVP mRNA, perhaps by displacing ribosomes or by slowing initiation or elongation, or through mediating sequestration into translationally inert storage or transport granules.

peptide production. Indeed, our results are consistent with previous reports that showed that Caprin-2 inhibits translation (*Shiina and Tokunaga, 2010*). Thus we suggest that association with Caprin-2 may inhibit the production of AVP peptide, perhaps by directly interfering with the translational apparatus, or through sequestration of AVP mRNA in translationally inert mRNP storage or transport granules (*Buchan, 2014*). Interestingly, polysome analysis revealed a heavy class of EDTA-resistant AVP mRNA containing fractions that might represent storage granules (*Murphy and Carter, 1990*). We also note that AVP transcripts are known to be transported to dendrites (*Mohr et al., 2002*) and to axons (*Murphy et al., 1989*), although in the latter case, the AVP RNAs are characterised by a short poly(A) tail that does not change in length following osmotic stimulation (*Murphy et al., 1989*).

The osmotic status-dependent modulation of AVP mRNA poly(A) tail length in the SON and PVN was first described over 20 years ago (*Carrazana et al., 1988*; *Zingg et al., 1988*; *Carter and Murphy, 1989*), but only now, with the discovery of Caprin-2, do we have insight into the regulation and physiological function of this post-transcriptional process. To get a more comprehensive picture of Caprin-2 functioning in the HNS, it will be of interest to identify other mRNAs that bind to Caprin2 in the SON and PVN, in addition to those encoding AVP. Further, identification of the sequences within the AVP mRNA that are recognised by Caprin-2, and the Caprin-2 protein modules that mediate binding to the AVP mRNA and poly(A) tail extension, might reveal interactions with other known Caprin-2 functions, such as the modulation of Wnt signaling (*Ding et al., 2008*; *Miao et al., 2014*; *Flores and Zhong, 2015*) that may have regulatory and physiological relevance.

## Materials and methods

### Animals and physiological manipulations

Male Sprague–Dawley rats (Harlan, UK) weighing ∼300 g were housed at a constant temperature of 22°C and a relative humidity of 50–60% (vol/vol) under a 14:10 hr light/dark cycle. Rats had free access to food and tap water for at least 1 week prior to experimentation. To induce hyperosmotic stress, water was removed for 3 days (dehydration, DH) or replaced by 2% (wt/vol) NaCl in drinking water for 7 days (salt loading, SL). The control group (euhydrated, EU) had access to food and water ad libitum throughout the experimental period. Rats were randomly allocated into groups. Food was available ad libitum. Tissue collections were performed between 10 am–2 pm. All experiments were carried out under the licensing arrangements of the UK Animals (Scientific Procedures) Act (1986) with local ethics committee approval.

### Tissue harvesting and RNA extraction

Rats were sacrificed by stunning and decapitation. Brains were removed and immediately frozen in powdered dry ice. SON and PVN samples were isolated from 60 μm frozen sections in a cryostat using 1 mm tissue punch tool (Fine Scientific Tools, Germany). The accuracy of the tissue punch was controlled by staining each slice with 2% (wt/vol) toluidine blue, and visualizing it on a light microscope. Samples were stored in −80°C until further analysis. Total RNA was extracted as described (*Greenwood et al., 2014*).

### Cloning and characterization of rat brain Caprin-2 mRNA isoforms

1 μg of RNA isolated from the rat forebrain was reverse transcribed using Super Script II RT kit (Life Technologies, Carlsbad, CA, USA). 2 μl of cDNA was used as a template for PCR with Caprin-2-specific primers rCAP2_F 5′ GCTCGAGGCCACCATGAAGTCAGCCAAGTCC 3′ and rCAP2_R 5′ GGGGATCGATTCAATCTTGATAAAGAAGATAGCC 3′, and Phusion High-Fidelity DNA Polymerase (New England Biolabs, Ipswich, MA, USA) under the following conditions: 98°C—2 min, (98°C—30 s, 53°C—30 s, 72°C—3 min) ×40, 72°C—10 min. The 3 kb PCR product was cloned using routine procedures, and sequenced with SP6 and T7 primers (Source BioScience, UK) as well as with Caprin-2-specific internal primers F 5′ GAAGGAACTTGTACAGCCAGA 3′ and R 5′ GATAAATGGCTGAG CAGGTC 3′ designed using OligoPerfect tool (Life Technologies). To identify different isoforms of Caprin-2 several different clones were analysed by restriction with frequently cutting enzymes (4-cutters): Alu I (Gibco, Life Technologies, USA), Dpn I (Stratagene, San Diego, CA, USA), Hae I and Rsa I (New England Biolabs), followed by agarose gel, and sequencing analysis (Source BioScience).

We found that, similar to mouse Caprin-2 (ENSMUST00000111569), the rat Caprin-2 gene consists of 18 coding exons (*Supplementary file 1*; Genbank accession number—KT867373). To maintain consistency with the mouse sequence, we labeled them 2–19. Since several different isoforms of Caprin-2 have been found in human (*Aerbajinai et al., 2004*), we also looked for different Caprin-2 isoforms in rat brain. We found that the longest isoform was 3090 bp long, which encodes 1029 aa. Exons were predicted on the basis of GT and AG flanking regions of introns in rat Caprin-2 gene (NCBI reference sequence AC_000072.1, 170273255–170328842, complement strand) and verified by comparison to the mouse Caprin-2 sequence ENSMUST00000111569. We then identified four other isoforms, which are shorter than the first one due to the absence of several base pairs (Cap2-2), or to the removal of entire exons (E13 and 14 in Cap2-3, E17 in Cap2-4, E13 in Cap2-5) (*Supplementary file 1*).

Previous studies have shown that human Caprin-2 is homologous to *Xenopus* RNG105, which is also known in rodents as Caprin 1 (*Shiina and Tokunaga, 2010*). The sequence of RNG105 has been thoroughly analyzed, therefore we used it as a reference to check if the rat Caprin-2 contains similar functional domains. Comparison of the rat 3090 bp Caprin-2 sequence to the *Xenopus* RNG105 revealed that they share several domains, including N-terminal coiled-coil domain thought to be involved in RNA binding, a nuclear localization signal, as well as a C-terminal RNA-binding motif, RGG (*Supplementary file 1*) also reported to be present in human Caprin-2.

## qRT-PCR

Steady state mRNA levels were assessed using quantitative real-time RT-PCR analysis. For cDNA synthesis, 100 ng of total RNA was treated with DNAse I (Invitrogen, Life Technologies, UK) and reverse transcribed using SuperScript III Reverse Transcriptase (Invitrogen Life Technologies) with random primers (Invitrogen Life Technologies), in the presence of RNaseOUT Inhibitor (Invitrogen Life Technologies). RNA isolated from cell cultures (100 ng) was reverse-transcribed using QuantiTect Reverse Transcription Kit (Qiagen, UK) according to the manufacturer protocol. Quantitative PCRs were conducted in duplicate, in 25 µl reaction volumes containing 1 or 2 µl of cDNA template, 0.4 µM gene-specific primers and FastStart Universal SYBR Green Master mix (ROX) (Roche, Switzerland), using ABI 7500 Real-Time PCR System (Applied Biosystems, UK). The following primers were used: Caprin-2 (F 5′ AGGTATCCAAGCCTGTGGTG 3′, R 5′ AGGATCTGCTGCCACTCTGT 3′), AVP (F 5′ TACGCTCTCTGCTTGCTTCC 3′, R 5′ ACTGTCTCAGCTCCATGTCG 3′), Rpl19 (ribosomal protein L19, F 5′ GTCCTCCGCTGTGGTAAAAA 3′, R 5′ GGCAGTACCCTTCCTCTTCC 3′), Gapdh (glyceraldehyde-3-phosphate dehydrogenase, F 5′ ATGATTCTACCCACGGCAAG 3′, R 5′ CTGGAA GATGGTGATGGGTT 3′ [*Colomer et al., 2010*]), eGFP (F 5′ ACTTCTTCAAGTCCGCCATGCC 3′, R 5′ TGAAGTCGATGCCCTTCAGCTC 3′), β-actin (F 5′ CACCCGCGAGTACAACCTTC 3′, R 5′ CCCA TACCCACCATCACACC 3′). Unless specified otherwise, primers were designed using either OligoPerfect (Invitrogen Life Technologies), MWG Operon (Eurofins Scientific, Luxembourg) or NCBI Primer Blast (*NCBI Resource Coordinators, 2015*) tools. Primers efficiency was validated using serial dilutions of cDNA from the SON or PVN. All qRT-PCR reactions were followed by dissociation curve analysis. Relative quantification of gene expression was performed using the $2^{\Delta\Delta CT}$ method (*Livak and Schmittgen, 2001*) and Rpl19 or GAPDH (in HEK cells) housekeeping genes.

## Immunohistochemistry

Rats were anesthetised with sodium pentobarbitone (100 mg/kg i.p.) and transcardially perfused with 0.1 M phosphate buffered saline (PBS, pH 7.4) followed by 4% (wt/vol) paraformaldehyde (PFA) in 0.1 M PBS. The brains were removed, post-fixed overnight in 4% (wt/vol) PFA, followed by 3 day-incubation in 30% (wt/vol) sucrose prepared in PBS, and frozen in liquid nitrogen. Coronal sections of the forebrain (30 µm) were cut on a cryostat, washed in 0.1 M PBS (pH 7.4) and subjected to antigen retrieval in 0.1 sodium citrate pH 6.0 at 100°C for 15 min. After three washes in PBS sections were blocked in 10% (vol/vol) donkey serum (Sigma–Aldrich, UK) in PBS supplemented with 0.3% Triton-X100 for 30 min. Floating sections were incubated with primary antibodies in 0.1 M PBS supplemented with 1% normal donkey serum, 0.3% (vol/vol) Triton X-100 for 1 hr at RT and then overnight at 4°C. The following primary antibodies were used: goat anti-Caprin-2 (Santa Cruz Biotechnology, Dallas, TX, USA), rabbit anti-Caprin-2 (kindly provided by Prof. Lin Li, Shanghai, China [*Ding et al., 2008*]), rabbit anti-GFP (Abcam, UK), and mouse anti-AVP-neurophysin (NP-II, PS41; 1: 100) (kindly provided by Prof. Harold Gainer, Bathesda, MD, USA [*Ben-Barak et al., 1985*]). Note that

similar results were obtained with both of the antibodies recognising Caprin-2. The data shown in *Figures 2, 3* were obtained using the goat anti-Caprin-2 from Santa Cruz Biotechnology.

After three washes in PBS (15 min each) sections were incubated for 1 hr at RT with a mixture of two or three fluorescent secondary antibodies: donkey anti-goat IgG—AF 488 or AF 594, donkey anti-mouse IgG—AF 594 or AF 647, donkey anti-rabbit IgG—AF 488 (1:500, Life Technologies), diluted 1:500 in PBS containing 1% (vol/vol) normal donkey serum and 0.3% (vol/vol) Triton X-100. At the end sections were incubated for 5 min at RT with DAPI (1 µg/ml in PBS, Sigma–Aldrich, St. Louis, MO, USA) and washed three times for 15 min, before mounting on slides. Fluorescent imaging was performed using either, Leica DM IRB epifluorescent microscope with Volocity (Improvision) acquisition system and cooled CCD camera (Hamamatsu ORCA ER Firewire) or Leica SP2 confocal microscope in the Wolfson Bioimaging Facility, University of Bristol. Fiji software (NIH, USA [*Schindelin et al., 2012*]) was used to quantify fluorescence intensities and co-localization of Caprin-2, AVP, GFP and DAPI in multichannel confocal image stacks.

## Construction of Caprin-2 shRNA and production of lentiviruses

The Caprin-2 shRNA was designed using BLOCK-iT RNAi Designer (https://rnaidesigner.life-technologies.com/rnaiexpress/; Life Technologies). A scrambled control shRNA for this oligonucle-otide was generated using siRNA Wizard v3.1 (www.sirnawizard.com/scrambled.php). We designed 4 different shRNAs, then tested them in HEK293 cells co-transfected with a Caprin-2 overexpression plasmid.

Only one shRNA delivered >55% knockdown (c2784, 40.51% $\pm$ 7.81 of control values) and was used in all further studies. Oligonucleotides (c2784 Caprin-2 shRNA, top strand: 5′ GGGAGAGACCTTTGATCTT CATTCAAGAGATGAAGATCAAAGGTCTCTCCCTTTTTT 3′, bottom strand: 5′ AATTAAAAAAGGGA GAGACCTTTGATCTTCATCTCTTGAATGAAGATCAAAGGTCTCTCCCGGCC 3′; scrambled Ctrl shRNA, top strand: 5′ GCGTTAAGCGAGCATGTTCTATTCAAGAGATAGAACATGCTCGCTTAACGCTTTTTT 3′, bottom strand: 5′ AATTAAAAAAGCGTTAAGCGAGCATGTTCTATCTCTTGAATAGAACATGCTCGCT TAACGCGGCC 3′) containing the loop sequence TTCAAGAGA were purchased from Eurofins MWG Operon, Germany. Double-stranded oligonucleotides were cloned into the pSilencer 1.0—U6 vector (Ambion, Life Technologies, Carlsbad, CA, USA). The efficiency of the specific Caprin-2 shRNA was tested in HEK293T/17 cells (human embryonic kidney cell line, CRL-11268, ATCC, Manassas, VA, USA) transiently overexpressing rat Caprin-2. The U6-shRNA sequences were then amplified from pSilencer 1.0-U6 (F 5′ ATAGATTTAATTAACACTATAGGGCGAATTGGGTA 3′, R 5′ AGTCTTTCTCGAGCCCGGGCTGCAG-GAATTA 3′) then cloned into LV pRRL.sin.U6.shRNA.cppt.CMV.GFP.wpre. High titer LVs were propagated as previously reported (*Greenwood et al., 2014*).

## Lentivirus-mediated Caprin-2 knockdown in the SON and PVN

Preliminary experiments were performed to assess the efficiency of Caprin-2 shRNA lentivirus transduction in vivo. Rats (300–340 g) were anaesthetized by i.m. administration of Domitor/Vetalar (Pfizer, New York City, NY, USA) and placed in a stereotaxic frame in the flat skull position. A 2 cm rostral-caudal incision was made to expose the surface of the skull. One 0.8 mm hole were drilled at co-ordinates 1.3 mm posterior to bregma and 1.95 mm left to the midline for SON injection. An additional one 1 mm hole was drilled at co-ordinates 1.8 mm posterior to bregma, and $\pm$ 0.4 mm lateral to midline for PVN injection. A 5 µl pulled glass pipette (Sigma–Aldrich) was positioned −8.8 mm (SON) or −7.5 mm (PVN) ventral to the surface of the brain and 1 µl of LV at the concentration of $2 \times 10^9$ particles/ml, was delivered separately into each nucleus, over 5 min/nucleus. At the end of surgery the rats received an agonist Antisedan (Norden Laboratories, Lincoln, NE, USA) and analgesic Rimadyl (Zoetis, Florham Park, NJ, USA). After 3 weeks the animals were deeply anaesthetized (Lethobarb, Fort Dodge, IA, USA) and perfused with PBS and PFA, as described before. Brain samples were further fixed in 4% (wt/vol) PFA for 24 hr, cryoprotected in 30% (wt/vol) sucrose in PBS and frozen in liquid nitrogen. Coronal hypothalamic sections were stained with anti-GFP (Abcam), AVP NP-II (PS-41, kindly provided by Prof. Harold Gainer, NIH, Bethesda, USA [*Ben-Barak et al., 1985*]) and Caprin-2 antibodies (Santa-Cruz Biotechnology) and the imaging was performed using fluorescent confocal microscope (Leica, Germany) as described earlier. Quantification analysis has been performed on the Z-stack and single-plane confocal microscope images obtained from the SON

and PVN of 5 rats injected with scrambled (Ctrl) or Caprin-2 (Cap2 KD) shRNA, using Fiji software (*Schindelin et al., 2012*).

Similar surgical procedure was performed for determination of physiological results of Caprin-2 knockdown, except that it involved bilateral injections into the SON and PVN. To this end two 0.8 mm holes were drilled at co-ordinates 1.3 mm posterior to bregma and 1.95 mm lateral to midline for SON injection and one 1 mm hole was drilled at co-ordinates 1.8 mm posterior to bregma, and ± 0.4 mm lateral to midline for PVN injection, and 1 µl of LV at the concentration of $2 \times 10^9$ particles/ml, was delivered separately into four nuclei, over 5 min/nucleus. Like previously, at the end of surgery the rats received Antisedan (Norden Laboratories) and analgesic Rimadyl (Zoetis). After the surgery the animals were individually housed in standard laboratory cages for two and a half week before being transferred to metabolic cages (Techniplast, Italy), for precise daily measurements of fluid intake and urine output. Animals were weighed and allowed to acclimatize to the cage for 48 hr. Fluid intake, urine output, urine osmolality and body weights were recorded for 10 days, between 10–11 am. The animals received ad libitum water for the first 3 days, and 2% (wt/vol) NaCl for the next 7 days. At the end of the experiment the animals were culled by stunning and decapitation and trunk blood samples were collected and processed immediately for plasma isolation. Brain tissue was frozen in powdered dry ice and stored in −80°C for RNA extraction and qRT-PCR and Northern blot analysis. Accuracy of injections and Caprin-2 knockdown were confirmed by eGFP and Caprin-2 mRNA expression analysis in each individual SON and PVN using qRT-PCR. Two separate groups of animals were used. In the first group, individual Caprin-2 shRNA-injected nuclei with 25% or higher Caprin-2 knockdown (as compared to the average control, scrambled shRNA-injected PVNs or SONs) were retrospectively chosen for gene expression analysis. In the second group, rats with more than 30% of Caprin-2 gene knockdown in both SON and PVN (5 out of the 10 animals injected) were retrospectively chosen for analysis of physiological parameters.

## Measurement of urine sodium and osmolality

Urine osmolality was measured in 100 µl of 10×-diluted samples by freezing point depression using a Roebling micro-osmometer (Camlab, UK). Urine sodium levels were determined in 2 ml of 10×-diluted samples using Cole–Parmer Sodium Combination Epoxy Body Electrode (Vernon Hills, IL, USA) and Jenway 3310 pH/mV meter (UK). The ionic strength of standards and samples was adjusted with 4 M $NH_4Cl$/4 M $NH_4OH$ added in 1:50 ratio. Sodium concentration in samples was calculated from the calibration curve representing potentials (mV) recorded for 0.1–100 mM NaCl standards.

## Determination of plasma osmolality and AVP concentration

Blood plasma was isolated from trunk blood collected after decapitation into chilled EDTA-treated plastic tubes (5 µl 0.5 M EDTA pH 8.0/1 ml blood) and centrifuged at 1600×g for 15 min at 4°C. 100 µl plasma samples were placed on ice and plasma osmolality was determined within 2 hr, as described before. The remaining plasma was aliquoted and stored in −80°C for AVP determination. AVP was extracted from 0.5 ml of plasma with acetone and petroleum ether and measured using arg8-Vasopressin EIA kit, according to the manufacturer protocol (Enzo Life Sciences, Farmingdale, NY, USA).

## RNA immunoprecipitation assay

Tissue punches from the left and right SON or PVNs were cross-linked with 1% (vol/vol) formaldehyde (Sigma–Aldrich) in PBS, for 10 min at room temperature. Cross-linking was terminated by 5 min incubation with 0.125 M glycine (pH 7.0) and samples were washed three times in PBS, using a centrifuge (5 min, 2000×g, 4°C). After homogenization in 100 µl of NT- RNA immunoprecipitation buffer (50 mM Tris, pH 7.4, 150 mM NaCl, 1 mM $MgCl_2$, 0.5% Nonidet P40, 1 mM EDTA pH 8.0, 1 mM DTT), Complete protease inhibitor (Roche), 200 U/ml RNase Out (Invitrogen, USA) samples were incubated for 10 min on ice and pre-cleared with 25 µl of Protein G-coated Dynabeads (Life Technologies). Protein levels were determined using BCA method (Thermo Fisher Scientific, Pierce, Waltham, MA, USA). First, we analyzed suitability of two different anti-Caprin-2 antibodies: rabbit (kindly provided by Prof. Lin Li, Shanghai, China) and goat (Santa Cruz Biotechnology), against non-specific rabbit IgG (Santa Cruz Biotechnology) and goat IgG specific to an irrelevant antigen DKK (Dickkopf-related protein 1, Santa Cruz Biotechnology). Both antibodies recognising Caprin-2 gave similar results. The data presented here was derived using the antibody raised in goat from Santa Cruz Biotechnology. Tissue extracts containing 60 µg (PVN samples) or 30 µg (SON samples) of protein

were incubated overnight on a rotating wheel at 4°C with either, goat anti-Caprin-2 antibody (Santa Cruz Biotechnology) or with non-specific goat IgG (Santa Cruz Biotechnology) at a concentration ratio 1:8, in PBS buffer supplemented with RNase Out (200 U/ml) and Complete Protease Inhibitor. Extracts equivalent to 20% of each sample were preserved in −80°C for input mRNA analysis. Next, the protein—antibody extracts were incubated with G-protein Dynabeads (pre-washed with PBS and blocked with 10% BSA for 10 min at RT) for 2 hr, at 4°C, on a rotating wheel. After 3 washes with PBS, the protein-G-adsorbed complexes were washed out with 50 µl of the crosslinking reversal buffer (50 mM Tris-HCl pH 7.0, 5 mM EDTA, 10 mM DTT, 200 U/ml RNase Out) at 70°C, for 45 min. All steps have been carried out in RNase-free environment, in a presence of RNase Out (washing buffers—50 U/ml, reaction buffers—200 U/ml) and Complete Protease Inhibitor (Roche). Total RNA was isolated with 1 ml of Reagent (Invitrogen, Life Technologies), cleaned with RNeasy MinElute Cleanup kit (Qiagen) and eluted with 14 µl of water, of which 8 µl was treated with DNase I (Life Technologies) and used for qRT-PCR analysis, as described above.

## Northern blot analysis of mRNA poly(A) tails

RNA extracted from SONs and PVNs of rats injected with lentiviruses and from the in vitro experiment on HEK cells was subjected to Northern blot analysis using Ambion NorthernMax kit (Life Technologies). Briefly, samples containing 300 ng of RNA were incubated for 15 min at 65°C with formaldehyde load dye containing ethidium bromide (10 µg/ml) and separated (5 V/cm) on 1% (wt/vol) agarose gel along with the BrightStar Biotinylated RNA Millenium marker (Life Technologies). The gel was exposed under UV light and photographed, then downward-transferred overnight to the BrightStar-Plus membrane and crosslinked for 2 min using a standard UV transilluminator. Quality of the transfer was assessed by visualizing the gel and the membrane under the UV light. Membrane was placed in the hybridization bag (Roche) and subjected to prehybridization, blocking and overnight hybridization at 37°C ULTRAhyb buffer. We used 100 pM oligonucleotide probes double biotinylated at the 5′ and the 3′ ends (Eurofins MWG Operon). The sequences of the anti-sense (AS) oligonucleotide probes used are as follows:

rAVP-AS–5′ GTAGACCCGGGGCTTGGCAGAATCCACGGACTCTTGTGTCCCAGCCAG 3′,
rGAPDH-AS–5′ CCAGCCTTCTCCATGGTGGTGAAGACGCCAGTAGACTCCACGACA 3′.

After washing in low stringency solution (2 × SSC, 0.1% wt/vol SDS), the signal was developed using the chemiluminescent BrightStart BioDetect kit (Ambion, Life Technologies, USA) and captured on Amersham Hyperfilm ECL (GE Healthcare, UK). At the end the probes were stripped by dipping membranes in boiling DEPC-treated water with 0.1% SDS until they reach RT, and re-probed for GAPDH.

To determine if the observed changes in the size of AVP mRNA were associated with changes in the length of their poly(A) tails, in parallel we performed Northern blot analysis on samples subjected to poly(A) tail removal. 200 ng of RNA was incubated with oligo(dT$_{12–18}$) primers (Life Technologies) for 5 min at 85°C to denature RNA, followed by 10 min hybridization at 42°C and slow cooling (1°C/min) to 32°C. Double-stranded poly(A) tails were then digested with 3.75 U of RNase H (New England Biolabs) at 37°C for 30 min. Samples were cleaned with 1 volume of phenol-chloroform, precipitated for 30 min at −20°C with 20 µg/ml of linear polyacrylamide (Ambion, Life Technologies), 10% (vol/vol) of 5 M ammonium acetate and 2 vol of 100% (vol/vol) ethanol and centrifuged at 12,000×g, 4°C for 20 min. After cleaning with ethanol the RNA pellet was resuspended in 20 µl of formaldehyde load dye containing ethidium bromide (10 µg/ml, Sigma–Aldrich), incubated for 15 min at 65°C and separated in agarose gel alongside the non-digested samples as described before. RNase-free conditions were maintained throughout the whole protocol. Molecular sizes of the RNA bands were calculated based on the migration distances of the ladder bands in each individual gel.

## In vitro studies

HEK 293T/17 cells (ATCC, USA) were plated into 6-well plates at a density of 300,000 cells/well in DMEM high glucose media (Sigma–Aldrich) supplemented with 10% (vol/vol) FBS, 2 mM L-glutamine and 1× NEAA (Sigma–Aldrich). The following day media was changed and cells were co-transfected with the rat genomic AVP structural gene under the transcriptional control of the CMV promoter (derived from pSP72-VP [*Chooi et al., 1994*]) in combination with either: (a) control pRRL.sin.cppt.CMV.eGFP.wpre vector ('eGFP'), (b) Caprin-2 overexpression vector pRRL.sin.cppt.CMV.rCaprin2.ires.eGFP.wpre ('Caprin-2'), (c) Caprin-2 overexpression vector and control scrambled shRNA vector

pRRL.sin.U6.Scr_shRNA.cppt.cmv.GFP.wpre ('Ctrl') and (d) Caprin-2 overexpression vector and Caprin-2 shRNA vector pRRL.sin.U6.Caprin-2_shRNA.cppt.CMV.GFP.wpre ('Cap2 shRNA'). Transfection with 2 µg of each plasmid and Lipofectamine LTX reagent (Life Technologies) was performed according to the manufacturer protocol. 48 hr after transfection cells were lysed with TRIzol reagent (Life Technologies) and subjected to RNA extraction, as described above.

## Western blotting

HEK293T/17 cells were cultured in 6 well-tissue culture plate for 24 hr (800,000 cells/wells). Transfections were performed in triplicate using standard calcium phosphate transfection method. Cells were either mock transfected, or were tranfected with control pRRL.sin.cppt.CMV.eGFP.wpre vector ('eGFP') or Caprin-2 overexpression vector pRRL.sin.cppt.CMV.rCaprin2.ires.eGFP.wpre ('Caprin-2'). Both vectors express eGFP. After 8 hr of transfection the culture medium was replaced with fresh media. At 2 days after transfection, total protein extraction was performed using RIPA buffer (1% vol/vol Nonidet P-40, 0.5% wt/vol sodium deoxycholate, 0.1% wt/vol SDS in phosphate buffer saline) containing protease inhibitors (P8340; Sigma). The lysate was incubated on ice for 30 min, followed by centrifugation at 10,000×$g$ for 10 min. Supernatant was collected and stored at −80°C. Protein concentration was determined using Bradford assay (Bio-Rad, Hercules, CA, USA). For immunoblot, proteins were separated by SDS-PAGE, then transferred to PVDF membranes (Millipore, Billerica, MA, USA). The membranes were blocked with 5% (vol/vol) ECL prime blocking reagent (Amersham, UK)/0.1% vol/vol tween20/TBS for 2 hr at room temperature and incubated with primary antibody diluted in 2% (vol/vol) ECL prime blocking reagent/0.1% vol/vol tween20/TBS at room temperature for at least 2 hr at room temperature or overnight at 4°C. Incubations with secondary antibodies conjugated with HRP were performed at room temperature for 1 hr. The signal was visualised using Westar EtaC HRP Detection Substrate (Cyanagen, Italy). Primary antibodies used were: goat polyclonal anti-Caprin2 (1:500, sc-107473; Santa Cruz Biotechnology), rabbit polyclonal anti-GFP (1:10,000, ab290; Abcam), mouse monoclonal anti-GAPDH (1:10,000, sc32233; Santa Cruz Biotechnology).

## Statistical analysis

Statistical analysis was performed using Graphpad Prism (Graphpad Software, La Jolla, CA, USA). Statistical differences between two groups in qRT-PCR, immunohistochemistry, Northern blot, RNA immunoprecipitation experiments and in plasma osmolality and AVP measurements were evaluated using unpaired Student's $t$-test. Results of qRT-PCR and immunofluorescence with three experimental groups were evaluated using one-way ANOVA with respectively, Holm-Sidak's and Bonferroni post-hoc tests. Two-way ANOVA with uncorrected Fisher's LSD test was used to determine the differences between more than two groups (i.e., urine output, osmolality and sodium concentration, fluid intake). $p < 0.05$ was considered significant. All data are expressed as the mean ± s.e.m.

## Acknowledgements

We gratefully acknowledge the support of the BBSRC (BB/G006156/1, AK, JP, DM; BB/J015415/1, JP, DM), a Wellcome Trust ISSF-Postdoctoral Research Staff Award (097822/Z/11/Z, AK), the University of Cardiff (AK), and High Impact Research Chancellory (UM.C/625/1/HIR/MOHE/MED/22 H-20001-E000086, DM) and Internationalisation Research (University of Malaya RP011-13HTM, S-YL, DM) grants from the University of Malaya.

## Additional information

### Funding

| Funder | Grant reference | Author |
| --- | --- | --- |
| Biotechnology and Biological Sciences Research Council (BBSRC) | BB/G006156/1 | Agnieszka Konopacka, Julian Paton, David Murphy |
| Biotechnology and Biological Sciences Research Council (BBSRC) | BB/J015415/1 | Julian Paton, David Murphy |

| Funder | Grant reference | Author |
|---|---|---|
| Wellcome Trust | 097822/Z/11/Z | Agnieszka Konopacka |
| Cardiff University | | Agnieszka Konopacka |
| Universiti Malaya | UM.C/625/1/HIR/MOHE/MED/22 H-20001-E000086 | Agnieszka Konopacka, Julian Paton, David Murphy |
| Universiti Malaya | RP011-13HTM | Su-Yi Loh, David Murphy |

The funders had no role in study design, data collection and interpretation, or the decision to submit the work for publication.

### Author contributions

AK, Conception and design, Acquisition of data, Analysis and interpretation of data, Drafting or revising the article; MG, S-YL, Acquisition of data, Analysis and interpretation of data; JP, DM, Conception and design, Analysis and interpretation of data, Drafting or revising the article

### Ethics

Animal experimentation: All experiments were carried out in strict accord with the licensing arrangements of the UK Animals (Scientific Procedures) Act (1986) (PIL30/2738). The protocols were also approved by the local Animal Welfare and Ethical Review Body (AWERB). Surgery was carried out under were anaesthesia by i.m. administration of Domitor/Vetalar, and every effort was made to minimise suffering.

## Additional files

### Supplementary file

• Supplementary file 1. The rat Caprin-2 gene. (a) The sequence of full-length rat brain Caprin-2 cDNA. (b) The predicted amino acid sequence of full-length rat brain Caprin-2 protein. (c) Alternatively spliced isoforms of rat brain Caprin-2. (d) Hypothetical functional domains of the rat brain Caprin-2 protein based on alignment to the *Xenopus* RNG105 protein sequence, a paralogue of the well-analyzed rat Caprin-1 protein, which is highly homologous to Caprin-2.

### Major dataset

The following dataset was generated:

| Author(s) | Year | Dataset title | Dataset ID and/or URL | Database, license, and accessibility information |
|---|---|---|---|---|
| Konopacka A, Murphy D | 2015 | Rat Caprin-2 | http://www.ncbi.nlm.nih.gov/nuccore/KT867373 | Publicly available at NCBI Nucleotide (accession no. KT867373). |

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
