## [Decision Letter]

Thank you for submitting your work entitled “RNA binding protein Caprin-2 is a pivotal regulator of the central osmotic defense response” for peer review at *eLife*. Your submission has been evaluated by James Manley (Senior Editor) and three reviewers, one of whom, Michael Green, is a member of our Board of Reviewing Editors.

The reviewers have discussed the reviews with one another and the Reviewing editor has drafted this decision to help you prepare a revised submission.

In this manuscript, Konopacka et al. identify a novel mechanism for the regulation of salt and water homeostasis by post-transcriptional control of the antidiuretic neurohormone arginine vasopressin (AVP). Previous studies by this lab have shown that Caprin-2 mRNA level is induced in response to osmotic challenges. In the present manuscript they show that the induction of Caprin-2 protein is important to maintain physiological responses to osmotic stimuli (i.e. fluid intake and urine output, osmolality and sodium concentration). They associate this response to the regulation of AVP levels by Caprin-2. Thus, they show that Caprin-2, which is a known RNA-binding protein, forms a complex with AVP mRNA following osmotic challenge. Through a set of experiments using shRNA knockdown and/or overexpression of Caprin-2 in vivo and in vitro they show that the binding of Caprin-2 to AVP affects the levels of AVP mRNA via regulation of AVP's poly(A) tail elongation.

The reinstatement of AVP levels in the hypothalamo-neurohypophyseal system (HNS) following chronic osmotic stimulation is critical for the organism's survival as AVP is the major regulator of water balance. Murphy and others previously reported that at the onset of an osmotic challenge AVP mRNA is subjected to an increase in the length of the poly(A) tail, which may affect AVP levels in neurons. In the present manuscript they provide a mechanism for this post-transcriptional modification and associate it to a critical physiological response.

Although the reviewers appreciate the importance of the work, they have raised a number of specific concerns that need to be addressed for publication in *eLife*. These are listed in the “Essential revisions” section. There are also a number of recommendations that the authors feel would strengthen the manuscript but are not essential and these are listed in the “Optional revisions” section.

Essential revisions:

1) It is standard in studies using siRNAs or shRNAs to rule out off target effects. This is typically done by showing that a second, unrelated siRNA or shRNA against the same target has a similar effect. The authors need to show that a second shRNA against Caprin-2, which is unrelated to the first Caprin-2 shRNA, has a similar effect in the key experiments/figures in this study. Alternatively, they can perform a “rescue” experiment involving ectopic expression of a Caprin-2 cDNA that is resistant to knockdown by the shRNA.

2) Figure 4: There needs to be some control for RNA binding specificity to rule out the possibility that Caprin 2 is a non-specific RNA binding protein. The most straightforward approach is to perform the RIP experiment with some other negative control RNA such as GAPDH.

3) Figure 1: Does the AVP antibody recognize AVP NP-II precursor peptide? If so, it should be labeled AVP NP-II, not just AVP. The experiments here should include the qPCR quantification of AVP mRNA along with Cap2 mRNA in SON and PVN isolated from EU, SL and DH rats (Figure 1). The immunostaining data in Figure 1 need to be quantified, so both protein and RNA levels of Cap2 and AVP in SON and PVN of EU, SL and DH rats can be compared side-by-side to provide a initial hint whether a change in translation efficiency may contribute to increased AVP synthesis. Alternatively, is it possible to divide the SON and PVN lysates to isolate RNAs for RT-qPCR and to prepare protein samples for western blotting? The images in Figure 1 seem to suggest that AVP-immunostained signal is weaker in SL than EU-stimulated SON. Moreover, osmotic stress-increased Cap2 but not AVP protein is more widely occurred in SON and PVN MCN neurons. Is this true? The authors claimed this is a “Caprin-2-like” signal. What does this mean? Are the authors not sure this signal is really Caprin-2? Some neurons with high AVP signal did not necessarily have high Cap2 signal, vice versa. Are Cap2 also up-expressed in non-AVP neurons in response to osmotic stress or some AVP secreted already?

In the figure legend, description of PVN (PVN 1.03 ± 0.04…) and SON (SON 1.01 ± 0.07…) data should be switched to follow the order of subfigures. For example, “Co-localization with AVP and OT” – what is OT? Oxytoxin? This abbreviation is also shown in Figure 4 but the entire manuscript does not address oxytoxin at all. In addition, scale bar of 100 μm is not marked in the Figure 1.

4) Figure 2: The control and Cap2 KD images are in different magnification without scale bars. Moreover, it is not clear why GFP is localized in the cytoplasm of Cap2 KD neurons but diffusely distributed in control neurons. Does “n” mean the number of animals? If so, why the number here is far more than the number of animals (Cap2 KD, n=5) used for physiology characterization in Figure 3? As the authors claimed, only the physiological data from rats with more than 30% of Cap2 KD in both SON and PVN were chosen for subsequent analysis. Does this mean that only 5 animals out of 11-26 (2a, n=11; 2c, n=26) rats injected with siCap2 lentivirus fit in this criteria? Here, we did not see a decrease or increase in AVP-immunostained signal in Cap2 KD neurons in 2b images. Is this true? The authors should quantify not just Cap2 but also AVP-immunostained signal.

5) Figure 4 is a misleading figure. The data only demonstrated that Cap2 is in complex with AVP RNA in the RIP assay. Does Cap2 also bind to OT RNA? Otherwise, please remove OT RNA in the legend. Because SL increases the amounts of Cap2 protein and AVP RNA and the amount of immunoprecipitated Cap2 proteins under EU and SL conditions was not shown to see whether “the effect of SL on AVP mRNA binding” in Figure 4 is indeed increased. Such an increase in 4B could be simply caused by more input AVP RNA and more precipitated Cap2 protein and has nothing to do with increased binding under SL condition. In addition, RIP data alone is insufficient to draw the conclusion “Cap2 binds to AVP RNA”. The content of data in this figure is too little to stand alone.

6) Figure 5: A large fraction of poly(A) tail is not influenced by altering Cap2 expression because the presence of oligo(dT) and RNase H further reduced the size of AVP RNA (5C, 5C'). Please specify the poly(A) tail length (looks like is more than 100-150-bp) that is not affected by Cap2.

7) In the Discussion, please clarify the sentence: “These observations may be explained by water retention resulting from the elevated plasma AVP levels found in the Caprin-2 shRNA-transduced rats.”

The authors explain that paradoxically increased plasma AVP could be due to increased osmolarity. If so, could authors directly measure the AVP content from posterior pituitary where MCN-synthesized AVP was first delivered to. It is still confusing whether Cap2 KD directly alters (increases or decreases) AVP peptide production and whether the aberrant physiological responses in SL-stimulated Cap2 KD rats could be explained by elevated AVP action?

Optional revisions:

1) Figure 6: Please provide western blot of Cap2 in KD and overexpression conditions. In addition, can the protein level of AVP be detected by nonapeptide ELISA (used in Figure 3) or immunofluorescence labeling by the antibody detecting AVP NP-II (used in Figure 2)?

2) Figure 7: Does the AVP RNA with shorter or longer poly(A) tail distributed differentially in the polysome profile? Moreover, this figure should be combined with Figure 6 and the protein levels of Cap2 and AVP should be measured by western blotting/ ELISA/ immunostaining (whichever is appropriate to detect Cap2 and AVP) and included in this figure. Can overexpression of an RNA binding-defective Cap2 mutant not affect poly(A) tail of AVP RNA?

3) If HEK 293 cells can recapitulate the relation of AVP RNA and Cap2 observed in SON and PVN, whether expression of Cap2 promotes polyadenylation or inhibits deadenylation of AVP RNA and whether AVP RNAs with long or short poly(A) tail distribute differently in polysomes could be simply tested in this system. The authors should consider the molecular approaches used to study polyadenylation, deadenylation and translation efficiency in cells to experimentally test these possibilities without extensive speculations in the Discussion section. This is important since the authors found a paradoxical increase in circulating AVP peptide (Figure 3) that is not in conceptual accordance with decreased poly(A) tail and stability of AVP RNA found in SON and PVN. Despite Bartel's genome-wide study did not show strong correlation between poly(A) tail (at the steady state) and translation efficiency, their results do not rule out that translation efficiency of any given RNA can be influenced by

ongoing polyadenylation or deadenylation events. For example, the authors (along with many published studies) observed the strong correlation between the change in poly(A) length of AVP RNA in vivo as a function of the degree of physiological responses by altering AVP expression. However, Cap2 is only able to affect a portion of poly(A) tail (∼20-30 bp out of ∼160 bp under KD condition) in response to osmotic stress. Thus, the change in AVP expression is likely influenced by polyadenylation or deadenylation event itself instead of poly(A) length.

4) The authors claimed that they are unable to distinguish the role of Cap2 between a process that promotes polyadenylation or one that prevents deadenylation. Because the authors can recapitulate Cap2 effect in the poly(A) length of AVP RNA in HEK293 cells, it is doable tasks to delineate the role of Cap2 in these two aspects of regulation.

5) Figure 3: The big puzzle here is that the increased plasma AVP (detecting the mature AVP nonapeptide by ELISA) in Cap2 KD SL rats did not evoke stronger physiological responses than control SL rats. Can this measurement be done using blood withdrawal from rats at SL1 and SL3 besides SL7? How about the expression levels of AVP and Cap2 in posterior pituitary since the earlier study showed that poly(A) elongation of AVP RNA is only in hypothalamus but not in posterior pituitary?

6) Related to point 2, it is important to at least roughly map the position of the caprin-2 binding site on the AVP mRNA. Most likely it will be near the 3' end. Mapping can be accomplished by performing RIP or CLIP using region specific primer-pairs and/or by in vitro RNA-protein binding experiments.

7) A schematic model that summarizes the main findings would be helpful.

8) The section “Cloning and characterization of rat brain Caprin-2 mRNA isoforms” interrupts the flow of the text. It is more appropriate for the Methods section.

9) Figures 1 and 2: Scale bars are missing.

10) Figure 2: The upper and lower panels (Control and shRNA) should be presented at the same magnification. Also, there is no annotation of the colors used to label Caprin and AVP.

11) Figure 4: Histograms “a” and “b” are confusing. It is not clear that “a” is Caprin-2 binding of AVP mRNA Vs. Ig control whereas “b” is Caprin-2 binding of AVP mRNA in ehuhydrated vs. salt loaded animals. We would consider omitting “a”. A schemata depicting the RNA pull-down procedure would be helpful.

12) The reviewers suggest changing the subheading “Caprin-2 regulates AVP…in vitro” to “Caprin-2 directly regulates AVPin vitro”

13) In the Discussion the authors write: “Salt loading increases the amount of AVP mRNA bound to Caprin-2. This can be attributed both to the increases in Caprin-2 protein levels and the abundance of AVP mRNAs.” One of the reviewers found this confusing because the increased Caprin-2 protein levels are responsible for the AVP mRNA abundance. Thus, this is one not two phenomena, and the word “both” is confusing.

---

## [Author Response]

Essential revisions:

1) It is standard in studies using siRNAs or shRNAs to rule out off target effects. This is typically done by showing that a second, unrelated siRNA or shRNA against the same target has a similar effect. The authors need to show that a second shRNA against Caprin-2, which is unrelated to the first Caprin-2 shRNA, has a similar effect in the key experiments/figures in this study. Alternatively, they can perform a “rescue” experiment involving ectopic expression of a Caprin-2 cDNA that is resistant to knockdown by the shRNA.

As is standard in such studies, we designed 4 different shRNAs and then tested them in HEK293 cells co-transfected with a Caprin-2 overexpression plasmid.

Only one shRNA delivered >55% knockdown (c2784, 40.51%±7.81 of control values) and was used in all further studies. The Methods now include this information (subsection “Construction of Caprin-2 shRNA and production of lentiviruses”).

We agree that the suggested experiments would have value. However, we were only able to derive a single suitable shRNA. Further, we are subject to time, cost and, most importantly, ethical constraints – the proposed studies would consume a large number of animals in addition to those already used. For these reasons, we confirmed our results in a recapitulated in vitro system.

*2)*
Figure 4*: There needs to be some control for RNA binding specificity to rule out the possibility that Caprin 2 is a non-specific RNA binding protein. The most straightforward approach is to perform the RIP experiment with some other negative control RNA such as GAPDH.*

We used Rpl19 mRNA as a negative control, and these data are now included in the new Figure 5.

*3)*
Figure 1*: Does the AVP antibody recognize AVP NP-II precursor peptide? If so, it should be labeled AVP NP-II, not just AVP.*

The AVP antibody does indeed recognize NP-II precursor peptide, which has now been clarified in the figure legend (now Figure 2).

*The experiments here should include the qPCR quantification of AVP mRNA along with Cap2 mRNA in SON and PVN isolated from EU, SL and DH rats (*Figure 1*).*

AVP mRNA quantification data is now included in Figure 1. We have also included data showing that the abundance of our standard “house-keeping” control, Rpl19 mRNA, does not alter following these stimuli.

*The immunostaining data in*
Figure 1
*need to be quantified, so both protein and RNA levels of Cap2 and AVP in SON and PVN of EU, SL and DH rats can be compared side-by-side to provide a initial hint whether a change in translation efficiency may contribute to increased AVP synthesis. Alternatively, is it possible to divide the SON and PVN lysates to isolate RNAs for RT-qPCR and to prepare protein samples for western blotting? The images in*
Figure 1
*seem to suggest that AVP-immunostained signal is weaker in SL than EU-stimulated SON. Moreover, osmotic stress-increased Cap2 but not AVP protein is more widely occurred in SON and PVN MCN neurons. Is this true?*

We have quantified the SON experiments as suggested, and these data are now included in Figure 2. Caprin-2 protein levels increase in the SON, in parallel with the observed increase in the Caprin-2 mRNA abundance. AVP levels decrease. Note, however, that the steady-state level of AVP in hypothalamic magnocellular neurones is not necessarily related to translation rate. As we describe in some detail in the Introduction, the AVP precursor is subject to axonal transport from cell bodies in the hypothalamus to terminals in the posterior pituitary, a process that is activated by the need to deliver mature peptide to the circulation following an osmotic stimulus. A statement to this effect has been added to the Discussion section.

The authors claimed this is a “Caprin-2-like” signal. What does this mean? Are the authors not sure this signal is really Caprin-2?

This overly cautious term has been removed. Note that we used two different antibodies recognizing Caprin-2, produced in two different species – rabbit and goat. The rabbit Ab (provided by Prof. Lin Li, Shanghai) was previously evaluated ([16]. Caprin-2 enhances canonical Wnt signaling through regulating LRP5/6 phosphorylation. J Cell Biol 182: 865–872). Both antibodies produced identical results.

Some neurons with high AVP signal did not necessarily have high Cap2 signal, vice versa. Are Cap2 also up-expressed in non-AVP neurons in response to osmotic stress or some AVP secreted already?

The referee is correct; other cell-type in the PVN and SON express Caprin-2 in addition to AVP cells. There may well be oxytocin cells, but this remains to be determined. The text has been modified accordingly (subsection “Caprin-2 expression is up-regulated by osmotic stress in rat AVP neurons”).

In the figure legend, description of PVN (PVN 1.03 ± 0.04…) and SON (SON 1.01 ± 0.07…) data should be switched to follow the order of subfigures.

This has been changed as suggested.

*For example, “Co-localization with AVP and OT” – what is OT? Oxytoxin? This abbreviation is also shown in*
Figure 4
*but the entire manuscript does not address oxytoxin at all.*

Apologies. This has been removed.

*In addition, scale bar of 100 μm is not marked in the*
Figure 1*.*

Scale bars are now included (now Figure 2).

*4)*
Figure 2*: The control and Cap2 KD images are in different magnification without scale bars.*

Scale bars now included. New images at the same magnifications have also been added.

Moreover, it is not clear why GFP is localized in the cytoplasm of Cap2 KD neurons but diffusely distributed in control neurons.

We have presented better images.

*Does “n” mean the number of animals? If so, why the number here is far more than the number of animals (Cap2 KD, n=5) used for physiology characterization in*
Figure 3*? As the authors claimed, only the physiological data from rats with more than 30% of Cap2 KD in both SON and PVN were chosen for subsequent analysis. Does this mean that only 5 animals out of 11-26 (2a, n=11; 2c, n=26) rats injected with siCap2 lentivirus fit in this criteria?*

It has been clarified in the Methods (subsection “Lentivirus-mediated Caprin-2 knockdown in the SON and PVN”) that Figures 3 and 4 represent data from 2 different batches of rats. The first batch was used for validation of the tool (now Figure 3) and the second for the physiological experiment, in which 5 out of 10 rats injected bilaterally to the SONs and PVNs with Caprin 2 shRNA lentivirus met the specified criteria and were chosen for the subsequent data analysis (now Figure 4).

Here, we did not see a decrease or increase in AVP-immunostained signal in Cap2 KD neurons in 2b images. Is this true? The authors should quantify not just Cap2 but also AVP-immunostained signal.

Better images have been provided. We have quantified AVP levels in Figure 2. However, please note that the steady-state level of AVP in hypothalamic magnocellular neurones is not only determined by translation rate. As we describe in some detail in the Introduction, the AVP precursor is subject to axonal transport from cell bodies in the hypothalamus to terminals in the posterior pituitary, a process that is activated by the need to deliver mature peptide to the circulation following an osmotic stimulus. A statement to this effect has been added to the Discussion section (sixth paragraph).

*5)*
Figure 4
*is a misleading figure. The data only demonstrated that Cap2 is in complex with AVP RNA in the RIP assay. Does Cap2 also bind to OT RNA? Otherwise, please remove OT RNA in the legend.*

Apologies. This has been removed.

*Because SL increases the amounts of Cap2 protein and AVP RNA and the amount of immunoprecipitated Cap2 proteins under EU and SL conditions was not shown to see whether “the effect of SL on AVP mRNA binding” in*
Figure 4
*is indeed increased. Such an increase in 4B could be simply caused by more input AVP RNA and more precipitated Cap2 protein and has nothing to do with increased binding under SL condition. In addition, RIP data alone is insufficient to draw the conclusion “Cap2 binds to AVP RNA”.*

We have modified the text to ensure that we are very clear that we are not suggesting that there is any change in Caprin-2 affinity or binding efficiency for the AVP mRNA following salt-loading. As accurately pointed out by the reviewers, and also discussed in the paper, increases in the amount of AVP mRNA bound to Caprin 2 (i.e. Caprin-2-AVP mRNA complexes) in SL conditions may result from increases in both Caprin 2 protein and AVP mRNA levels.

The content of data in this figure is too little to stand alone.

The figure has been altered as suggested by the referees.

*6)*
Figure 5*: A large fraction of poly(A) tail is not influenced by altering Cap2 expression because the presence of oligo(dT) and RNase H further reduced the size of AVP RNA (5C, 5C'). Please specify the poly(A) tail length (looks like is more than 100-150-bp) that is not affected by Cap2.*

Please note that we suggest that Caprin-2 is responsible for extending the length of the poly(A) tail, not synthesising the entirety of the poly(A) tail. Thus, under salt-loaded conditions, the poly(A) returns to the length seen in euhydrated rats when Caprin-2 is knocked down. This has been clarified in the text (subsection “Caprin-2 regulates the length of AVP mRNA poly(A) tails in vivo”).

7) In the Discussion, please clarify the sentence: “These observations may be explained by water retention resulting from the elevated plasma AVP levels found in the Caprin-2 shRNA-transduced rats.”

The authors explain that paradoxically increased plasma AVP could be due to increased osmolarity. If so, could authors directly measure the AVP content from posterior pituitary where MCN-synthesized AVP was first delivered to. It is still confusing whether Cap2 KD directly alters (increases or decreases) AVP peptide production and whether the aberrant physiological responses in SL-stimulated Cap2 KD rats could be explained by elevated AVP action?

As we discuss extensively, we suggest that the physiological effects seen in Caprin-2 knockdown rats could be a consequence of elevated AVP action. Indeed, we see more circulating AVP in these animals. The question of the effect on Caprin-2 on AVP mRNA translation, whilst discussed, is beyond the scope of this paper.

Optional revisions:

*1)*
Figure 6*: Please provide western blot of Cap2 in KD and overexpression conditions. In addition, can the protein level of AVP be detected by nonapeptide ELISA (used in*
Figure 3*) or immunofluorescence labeling by the antibody detecting AVP NP-II (used in*
Figure 2*)?*

A Western blot showing Caprin-2 protein expression in transfected cells is shown in a Figure 7, which also diagrammatically conceptualises the experiment.

*2)*
Figure 7*: Does the AVP RNA with shorter or longer poly(A) tail distributed differentially in the polysome profile?*

This statement has been added to the text (Discussion, sixth paragraph):

“We have previously examined the polysome distribution of the AVP mRNA in euhydrated and salt-loaded SON9, and showed no difference in the pattern association with heavy polysome fractions. However, the AVP mRNA is small, and subject to a high rate of translation, even in euhydrated animals. Thus, the rates of translation initiation, elongation and termination, none of which are directly measured by polysome analysis, especially when the size of the RNA limits ribosome number, could be influenced, positively or negatively, by poly(A) tail length.”

*Moreover, this figure should be combined with*
Figure 6
*and the protein levels of Cap2 and AVP should be measured by western blotting/ ELISA/ immunostaining (whichever is appropriate to detect Cap2 and AVP) and included in this figure.*

See new Figure 7.

Can overexpression of an RNA binding-defective Cap2 mutant not affect poly(A) tail of AVP RNA?

This would be a most interesting experiment, but is beyond the scope of this manuscript. Hopefully we will soon secure funding to explore the mechanisms of Caprin-2 action in closer molecular detail.

3) If HEK 293 cells can recapitulate the relation of AVP RNA and Cap2 observed in SON and PVN, whether expression of Cap2 promotes polyadenylation or inhibits deadenylation of AVP RNA and whether AVP RNAs with long or short poly(A) tail distribute differently in polysomes could be simply tested in this system.

See above.

*The authors should consider the molecular approaches used to study polyadenylation, deadenylation and translation efficiency in cells to experimentally test these possibilities without extensive speculations in the Discussion section. This is important since the authors found a paradoxical increase in circulating AVP peptide (*Figure 3*) that is not in conceptual accordance with decreased poly(A) tail and stability of AVP RNA found in SON and PVN. Despite Bartel's genome-wide study did not show strong correlation between poly(A) tail (at the steady state) and translation efficiency, their results do not rule out that translation efficiency of any given RNA can be influenced by ongoing polyadenylation or deadenylation events. For example, the authors (along with many published studies) observed the strong correlation between the change in poly(A) length of AVP RNA in vivo as a function of the degree of physiological responses by altering AVP expression. However, Cap2 is only able to affect a portion of poly(A) tail (∼20-30 bp out of ∼160 bp under KD condition) in response to osmotic stress. Thus, the change in AVP expression is likely influenced by polyadenylation or deadenylation event itself instead of poly(A) length.*

Please note that we suggest that Caprin-2 is responsible for extending the length of the poly(A) tail, not synthesising the entirety of the poly(A) tail. Thus, under salt-loaded conditions, the poly(A) returns to the length seen in euhydrated rats when Caprin-2 is knocked down. This has been clarified in the text (subsection “Caprin-2 regulates the length of AVP mRNA poly(A) tails in vivo”).

4) The authors claimed that they are unable to distinguish the role of Cap2 between a process that promotes polyadenylation or one that prevents deadenylation. Because the authors can recapitulate Cap2 effect in the poly(A) length of AVP RNA in HEK293 cells, it is doable tasks to delineate the role of Cap2 in these two aspects of regulation.

These are interesting experiments, but are beyond the scope of this manuscript.

*5)*
Figure 3*: The big puzzle here is that the increased plasma AVP (detecting the mature AVP nonapeptide by ELISA) in Cap2 KD SL rats did not evoke stronger physiological responses than control SL rats. Can this measurement be done using blood withdrawal from rats at SL1 and SL3 besides SL7? How about the expression levels of AVP and Cap2 in posterior pituitary since the earlier study showed that poly(A) elongation of AVP RNA is only in hypothalamus but not in posterior pituitary?*

These are interesting experiments, but, again, they are beyond the scope of this manuscript.

6) Related to point 2, it is important to at least roughly map the position of the caprin-2 binding site on the AVP mRNA. Most likely it will be near the 3' end. Mapping can be accomplished by performing RIP or CLIP using region specific primer-pairs and/or by in vitro RNA-protein binding experiments.

Although interesting, these experiments are also beyond the scope of this paper.

7) A schematic model that summarizes the main findings would be helpful.

A schematic model has been included (new Figure 10).

8) The section “Cloning and characterization of rat brain Caprin-2 mRNA isoforms” interrupts the flow of the text. It is more appropriate for the Methods section.

This has been moved as suggested.

*9)*
Figures 1 and 2*: Scale bars are missing.*

This has been corrected.

*10)*
Figure 2*: The upper and lower panels (Control and shRNA) should be presented at the same magnification. Also, there is no annotation of the colors used to label Caprin and AVP.*

Better pictures at the same magnification have been provided, and annotations explained in the figure legend (now Figure 3).

*11)*
Figure 4*: Histograms “a” and “b” are confusing. It is not clear that “a” is Caprin-2 binding of AVP mRNA Vs. Ig control whereas “b” is Caprin-2 binding of AVP mRNA in ehuhydrated vs. salt loaded animals. We would consider omitting “a”. A schemata depicting the RNA pull-down procedure would be helpful.*

We have omitted (a) at the suggestion of the referee. RNA pull-down is a standard procedure that is described in detail in the Materials and methods. A schematic is now included in Figure 5 as suggested.

12) The reviewers suggest changing the subheading “Caprin-2 regulates AVP… in vitro” to “Caprin-2 directly regulates AVP… in vitro”

Thank you. This has been done.

13) In the Discussion the authors write: “Salt loading increases the amount of AVP mRNA bound to Caprin-2. This can be attributed both to the increases in Caprin-2 protein levels and the abundance of AVP mRNAs.” One of the reviewers found this confusing because the increased Caprin-2 protein levels are responsible for the AVP mRNA abundance. Thus, this is one not two phenomena, and the word “both” is confusing.

The referee makes a good point. This text has been cut so as to avoid confusion.